# Dissecting Bit-Level Scaling Laws in Quantizing Vision Generative Models

## Abstract

Vision generative models have recently made significant advancements along two primary paradigms: diffusion-style and language-style, both of which have demonstrated excellent scaling laws. Quantization is crucial for efficiently deploying these models, as it reduces memory and computation costs. In this work, we systematically investigate the impact of quantization on these two paradigms. Surprisingly, despite achieving comparable performance in full precision, language-style models consistently outperform diffusion-style models across various quantization settings. This observation suggests that language-style models have superior bit-level scaling laws, offering a better tradeoff between model quality and total bits. To dissect this phenomenon, we conduct extensive experiments and find that the primary reason is the discrete representation space of language-style models, which is more tolerant of information loss during quantization. Furthermore, our analysis indicates that improving the bit-level scaling law of quantized vision generative models is challenging, with model distillation identified as a highly effective approach. Specifically, we propose TopKLD to optimize the transfer of distilled knowledge by balancing "implicit knowledge" and "explicit knowledge" during the distillation process. This approach elevates the bit-level scaling laws by one level across both integer and floating-point quantization settings.

## 1 Introduction

Visual generative models have recently progressed rapidly along two primary trajectories. On the one hand, diffusion-style models (Ho et al., 2020; Dhariwal & Nichol, 2021; Sohl-Dickstein et al., 2015; Song & Ermon, 2019) have achieved significant success in various applications, such as text-to-image generation (Halgren et al., 2004; Ramesh et al., 2022), image editing (Kawar et al., 2023; Brooks et al., 2023; Meng et al., 2021), and image-to-image translation (Choi et al., 2021; Zhang & Chen, 2022), demonstrating impressive scaling laws, as evidenced by models like DIT (Peebles & Xie, 2023). On the other hand, motivated by the potential of visual tokenizers (Van Den Oord et al., 2017) and the success of large language models, language-style generative models have also seen substantial advancements (Razavi et al., 2019b; Yu et al., 2021). Pioneering efforts such as VQGAN (Esser et al., 2021b) and DALL-E (Ramesh et al., 2021) along with their successors, have demonstrated the potential of language-style models in image generation. The recent development of VAR (Tian et al., 2024) further underscores the effectiveness of this approach in exhibiting excellent scaling laws.

As visual generative models scale, the increasing number of parameters poses significant challenges in terms of memory footprint and inference latency. To mitigate these challenges, quantization has become a crucial technique, traditionally trading off accuracy for efficiency in a specific model (Xiao et al., 2023; Li et al., 2024c; Liu et al., 2024b). However, with the emergence of models in varying sizes, quantization must now optimize across different sizes and bit settings to maximize both performance and efficiency with a series of models. For example, a 4-bit 6B model often outperforms an 8-bit 3B model, even with the same total bit budget (Zeng et al., 2022). Bit-level scaling laws (Dettmers & Zettlemoyer, 2023) have become critical in predicting model performance, helping identify the best precision settings and quantization strategies to enhance accuracy while minimizing resource usage. Thus, the goal of quantization is shifting toward improving bit-level scaling laws to optimize the balance between efficiency and performance.

A key question arises: Do diffusion-style and language-style models exhibit similar bit-level scaling laws? To investigate this, we choose two representative model series, DiT and VAR for diffusion style and language style respectively, to exhibit their corresponding scaling laws. Our study covers models with parameters ranging from 300M to 7B and quantization levels from 3 to 16 bits. Furthermore, we conduct experiments using both post-training quantization (PTQ) and quantization-aware training (QAT) techniques (Nagel et al., 2021) under both weight-only and weight-activation quantization settings.

Our results indicate that while both types of models achieve comparable accuracy at full precision, language-style models consistently outperform diffusion-style models across various quantization settings. Specifically, the language-style model demonstrates better bit-level scaling laws than full precision, whereas diffusion style could even show worse scaling behaviors compared to full precision. Reducing the weight precision of the language-style model from 16 bits to 4 bits and the activation precision from 16 bits to 8 bits significantly enhances its generative performance compared to the full-precision W16A16 model, given the same memory and computing cost constraints.

To reveal the reasons for these differences, we analyze the generation process of the two model types. We contrast their tolerance to single-step and multi-step inference errors during quantization in Section 3.2. The results suggest that the discrete representation space introduced by the codebook in language-style generative models enhances their robustness to quantization, mitigating the impact of quantization noise on the final image quality. Moreover, we analyze the distribution of activation values across different layers during the inference process, as illustrated in Figure 4, the use of a consistent codebook as input features over time helps to alleviate the issue of high variance in activation features encountered in diffusion-style models, thereby providing a robust foundation for improved bit-level scaling laws.

Further, we observe that the language-style model's scaling behavior degrades when the weight precision is reduced to 3 bits. To improve the bit-level scaling of language-style generative models, we explored existing state-of-the-art quantization algorithms. Unfortunately, our observations indicate that they offered limited enhancement, with distillation methods only partially recover the scaling laws at lower bit precision, approximating the scaling behaviors of W4A16 and W8A8.

Based on our insights on the role of the codebook in representation space reconstruction, we propose the *TopKLD* method, which builds upon the inherent top-k sampling mechanism of the codebook to optimize knowledge transfer efficiency by balancing "implicit" and "explicit" knowledge, thereby facilitating advanced bit-level scaling behaviors. Notably, in scenarios where only weights are quantized, 3-bit models outperform those with 4-bit precision in terms of bit-level scaling behavior. Furthermore, under weight-activation quantization conditions, this approach allows W4A8 models to surpass the bit-level scaling performance of W8A8 models. We also examine the impact of data types on model scaling behavior. While data type variations can enhance bit-level scaling laws to a certain extent, they still exhibit similar trends to those seen in integer quantization. To further improve the model's bit-level scaling laws, we apply TopKLD distillation to floating-point quantization, which results in a more significant enhancement of the model's scaling performance.

The contributions of this paper are summarized as follows:

- We conducted a comprehensive analysis of existing visual generative models from bit-level scaling laws and found that, despite achieving comparable performance at full precision, language-style models consistently outperform diffusion-style models across various quantization settings.

- We uncover that the discrete representation space in language-style models significantly enhances their robustness to quantization. This robustness mitigates the effects of quantization noise, leading to better bit-level scaling laws.

- We propose the TopKLD-based distillation method, which balances the "implicit knowledge" and "explicit knowledge" derived from full-precision models, enhancing the bit-level scaling behaviors of language-style models by one level.

## 2 BACKGROUND

### 2.1 VISUAL GENERATIVE MODELS

Visual generative models are predominantly classified into two categories: diffusion-style models and language-style models.

**Diffusion-style generative models** Diffusion-style models (Sohl-Dickstein et al., 2015; Song & Ermon, 2019) are regarded as the state-of-the-art in visual generation due to their high-quality image (Saharia et al., 2022b; Rombach et al., 2022b) and video (Ho et al., 2022a; Saharia et al., 2022a; Blattmann et al., 2023a;b) generation, generating images by iteratively refining noisy inputs through a denoising process. The model learns a parameterized denoising function $p_\theta(x_t|x_{t+1})$ over the latent space, which can be summarized as the following process:

$$p_\theta(x_t|x_{t+1}) = \mathcal{N}(x_t; \frac{1}{\alpha_{t+1}}(\bar{\beta}_{t+1} - \alpha_{t+1}\sqrt{\bar{\beta}_t^2 - \sigma_{t+1}^2})\epsilon_\theta(x_{t+1}, t+1), \sigma_{t+1}^2 I) \tag{1}$$

where $x_t$ represents the noisy data at timestep $t$, when $\sigma_t = \frac{\bar{\beta}_{t-1}\beta_t}{\bar{\beta}_t}$, $\beta_t = \sqrt{1 - \alpha_t^2}$, it represents a standard diffusion process (Ho et al., 2020), whereas when $\sigma_t = 0$, the diffusion process from $x_t$ to $x_{t-1}$ is a deterministic transformation (Song et al., 2020). However, regardless of the specific diffusion process, diffusion-style models can generally be viewed as operating within a continuous latent space. They start with pure Gaussian noise $x_T$ and iteratively refine it through a denoising process to generate a high-quality image $x_0$. Moreover, numerous recent works have established a strong connection between diffusion models and the mathematical fields of stochastic differential equations (SDEs) and ordinary differential equations (ODEs) to optimize the diffusion generation process (Jolicoeur-Martineau et al., 2021; Liu et al., 2022; Lu et al., 2022a;b; Zheng et al., 2023). Among these, DiT (Peebles & Xie, 2023) has demonstrated promising scaling results, outperforming all prior diffusion models.

**Language-style generative models** Language-style models are conceptually derived from autoregressive approaches commonly used in natural language processing (NLP), where images are generated by predicting each element in a sequence based on the tokens previously generated. Specifically, motivated by the potential of visual tokenizer, language-style models utilize a visual tokenizer $f$ to transform visual inputs into sequences of discrete tokens. Given an image $V$ (where $T$ represents the batchsize of samples, $H$ is the height, and $W$ is the width), the visual tokenizer generates a discrete representation as follows:

$$X = f(V) \in \{1, 2, ..., K\}^{B' \times H' \times W'} \tag{2}$$

where $K$ denotes the size of the codebook (vocabulary), and $B', H', W'$ are the dimensions of the tokenized representation. The resulting discrete token representation $X$ is then reshaped into a sequence and input into a Transformer-based language model (LM) for generative modeling. In models like DALL-E, MAGVIT, and Parti, the goal is to predict each token $x_i$ conditioned on the preceding tokens and any additional context $c$ by modeling the conditional distribution:

$$p(x_1, x_2, ..., x_k) = \prod_{k=1}^{K} p(x_k|x_1, x_2, ..., x_{k-1}; c) \tag{3}$$

During inference, language-style models, which are based on autoregressive methods, adopt various decoding strategies. Models like ImageGPT (Chen et al., 2020), DALL-E (Ramesh et al., 2021), and Parti (Yu et al., 2022) utilize a GPT-style autoregressive approach to sequentially generate tokens. In contrast, models such as MaskGIT (Chang et al., 2022), MAGVIT (Yu et al., 2023a), Phenaki (Villegas et al., 2022), and MUSE (Chang et al., 2023) follow a BERT-style masked regression strategy (Yu et al., 2023b), generating tokens in parallel batches. While language-style models have historically lagged behind diffusion models in visual generation tasks, recent advancements have revitalized their potential. Among them, VAR (Tian et al., 2024) has demonstrated superior performance and has exhibited impressive scaling laws as well.

## 2.2 QUANTIZATION AND BIT-LEVEL SCALING LAWS

Quantization, a pivotal stage in model deployment, has often been scrutinized for its ability to reduce memory footprints and inference latencies. Typically, its quantizer $Q(X|b)$ is defined as follows:

$$Q(X|b) = \text{clip}(\left\lfloor \frac{X}{s} \right\rceil + z, 0, 2^b - 1) \qquad (4)$$

Where $s$ (scale) and $z$ (zero-point) are quantization parameters determined by the lower bound $l$ and the upper bound $u$ of $X$, which are usually defined as follow:

$$l = \min(X), u = \max(X) \qquad (5)$$

$$s = \frac{u - l}{2^b - 1}, z = \text{clip}(\left\lfloor -\frac{l}{s} \right\rceil + z, 0, 2^b - 1) \qquad (6)$$

Bit-level scaling law is a strong predictor of model performance. It facilitates the optimization of accuracy and efficiency by identifying optimal precision settings and quantization strategies within constrained bit budgets. An effective bit-level scaling law can achieve optimal performance while minimizing resource consumption. Early studies (Hestness et al., 2017; Rosenfeld et al., 2019; Kaplan et al., 2020) on LLMs scaling highlighted the need to understand how different variables evolve with scale, demonstrating that small block sizes and floating-point data types offer advantages in scaling efficiency (Zeng et al., 2022; Dettmers & Zettlemoyer, 2023). These studies revealed that leveraging unique scaling properties could maintain nearly identical performance even with low-bit quantization, without requiring post-training adjustments. However, the exploration of bit-level scaling laws within visual generative model quantization (Yuan et al., 2022; Li et al., 2022; 2023c;d) remains limited, our study is an essential step towards understanding how various models and quantization methods influence bit-level scaling behaviors.

## 3 EXPERIMENTAL & ANALYSIS

In our experimental study, we evaluate the VAR and DiT models on the ImageNet 256×256 (Deng et al., 2009) conditional generation benchmarks. The VAR series models include sizes of 310M, 600M, 1B, and 2B, while the DiT series models comprise 458M, 675M, 3B, and 7B. We investigate two quantization settings: weight-only quantization and weight-activation quantization, analyzing the scaling behaviors under fixed total model bits $\mathcal{MT}$ and total compute bits $\mathcal{CT}$ conditions.

Total model bits refer to the bit memory occupied by all weight parameters, reflecting the impact of weight memory in memory-bound scenarios, whereas total compute bits account for quantization effects on matrix computations, defined as the total bit memory of weights and activations involved. for a 7B model under W8A8 quantization, $\mathcal{MT} \propto 8$ (calculated as $8 \times 7 \times 10^9$), $\mathcal{CT} \propto 8^2$ (calculated as $8^2 \times 7 \times 10^9$). The quantization precision for weights is varied from 8 bits to 3 bits, while activation precision is varied from 16 bits to 8 bits, including configurations such as W3A16, W4A16, W8A16, W4A8, and W8A8.

Through our analysis, we observed that the Fréchet Inception Distance (FID) scores of the generative models followed a distinct bivariate power function with respect to both the number of parameters and the bit-precision levels. Notably, different bit-precisions exhibited nearly parallel scaling trends, thereby validating our decision to employ power laws to characterize these scaling behaviors.

### 3.1 WHICH TYPE OF VISUAL GENERATIVE MODELS DEMONSTRATE SUPERIOR BIT-LEVEL SCALING PROPERTIES?

We conducted an analysis of both model types using standard PTQ and QAT methods, with the results shown in Figure 1. Our findings reveal that, irrespective of the quantization method employed or whether only weights or both weights and activations are quantized, language-style models demonstrate superior bit-level scaling behavior. Furthermore, it is evident that within language-style models, the optimal bit-level scaling behavior is achieved with a 4-bit weight quantization when solely quantizing weights. Conversely, when both weights and activations are quantized, the W8A8 configuration provides the best scaling performance. Reducing the weight precision to 4-bit in this scenario results in a degradation of the model's scaling capabilities.

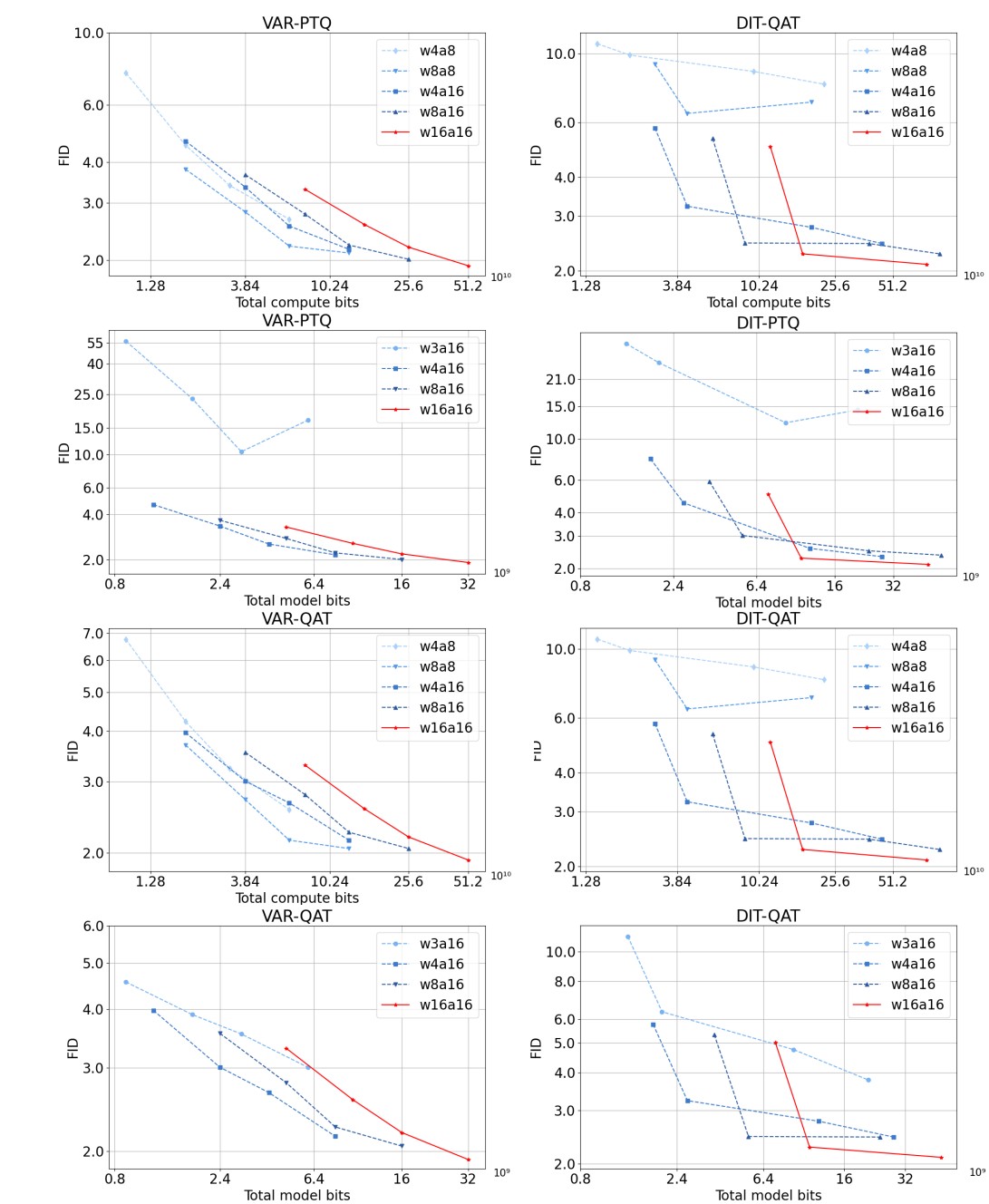

Figure 1: Investigation of bit-level scaling laws for VAR (left) and DiT (right) models using standard PTQ and QAT. Left: Quantied VAR exhibits better bit-level scaling laws than full-precision VAR (a shift towards the lower-left region). Right: Quantized DiT shows "almost" no improvement compared to full precision.

## 3.2 WHY DO LANGUAGE-STYLE GENERATIVE MODELS HAVE BETTER BIT-LEVEL SCALING LAWS?

Both types of generative models require multiple inference steps to produce the final image. To uncover the observed differences, we abstracted the inference processes of these models into two primary phases: model feature extraction and representation space reconstruction. This generative process is illustrated in Figure.2a.

We separately analyzed the errors after each stage. The representation reconstruction error directly reflects the generation quality of the models. Therefore, reducing error propagation during the generation process significantly improves the final output quality. As shown in Figure 2, our analysis reveals that language-style models exhibit higher fault tolerance in representation space reconstruction. We believe this is mainly because, compared to the continuous space of diffusion-style models, the reconstruction process of language-style models occurs in a discrete space, which can significantly absorb minor errors caused by quantization, resulting in greater resistance to interference.

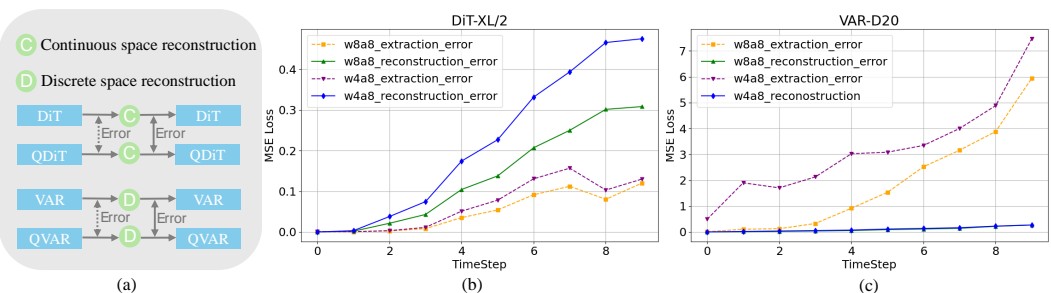

Figure 2: (a) denotes the generation process of visual generative models. A comparison of time-varying errors in quantized DiT (b) and VAR (c) indicates that, despite the errors introduced by quantization during the feature extraction phase in VAR, reconstruction significantly reduces these errors. Conversely, DiT fails to mitigate its errors and experiences an increase, adversely affecting the quality of the final output.

The above experiment qualitatively demonstrates that discrete spaces are more tolerant of information loss during quantization. To quantitatively validate this, we simulated the impact of quantization on feature extraction results by controlling Gaussian noise intensity via SNR (Box, 1988) and examined its effect during representation reconstruction. We designed two experiments to study both single-step quantization error and multi-step error accumulation.

**Tolerance to single-step quantization errors** We incrementally increased single-step noise intensity and compared the final generation quality to the original results. As illustrated in Figure 3a, the VAR model's loss exhibited a clear step-like progression, highlighting its discrete space's fault tolerance. In contrast, the correlation coefficient (Sedgwick, 2014) between loss and noise intensity for DiT (0.99) was significantly higher than for VAR (0.86), indicating DiT's continuous representation space is more sensitive to errors.

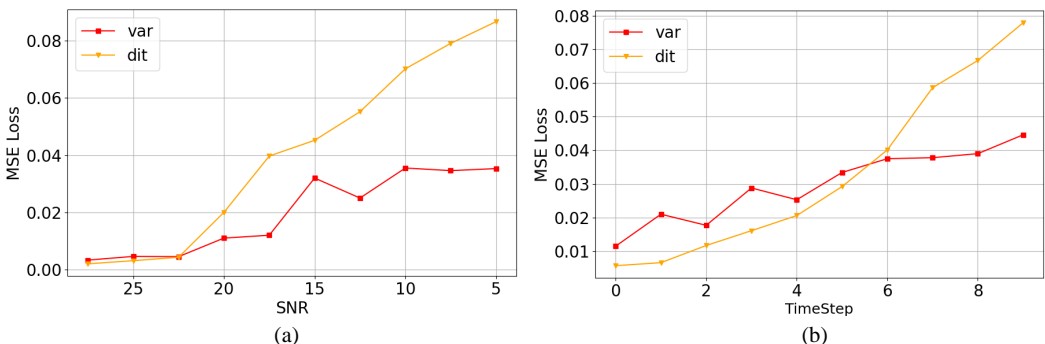

Figure 3: Analysis of Fault Tolerance in Representation Space Reconstruction Errors. (A lower SNR indicates a higher noise component).

**Tolerance to accumulated quantization errors across multiple steps** Since both diffusion-style and language-style models rely on multiple inference steps to generate final results, we analyzed the error accumulation from quantization during the reconstruction process. Specifically, equal-intensity noise is introduced in the initial 10% of inference steps, while the remaining 90% are noise-free. As shown in Figure 3b. Our findings reveal that the diffusion-style model displayed significant

error accumulation during the reconstruction process, whereas the language-style model showed a fluctuating increase in error. This behavior can be attributed to the fault tolerance inherent in the discrete representation space of language-style models, which mitigates the impact of quantization errors introduced during the early stages of inference.

**Activation distribution** Finally, we analyzed the activations of both models, with the visualization results shown in the Figure 4. It is observed that the variance of activations over time in the VAR model did not exhibit the same pronounced fluctuations as in the DiT model. This significantly reduces the difficulty associated with quantizing activations.

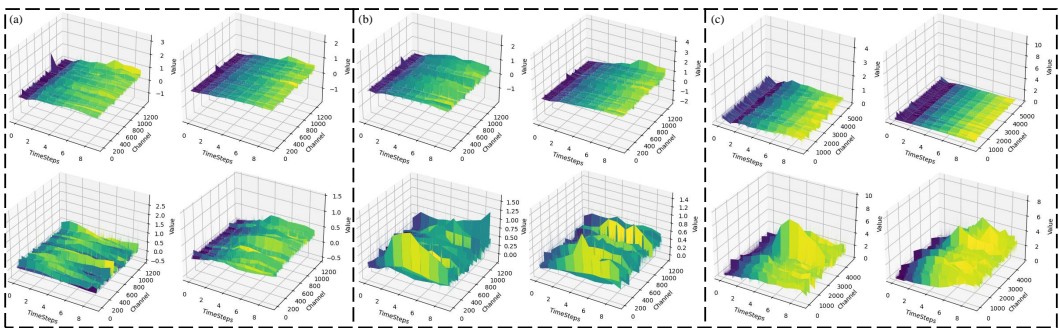

Figure 4: Visualization of activation values in the 5th, 15th transformer blocks for VAR (top) and DiT (bottom), focusing on the FC1, QKV, and FC2 layers. For additional visualization results, please refer to the appendix B.

### 3.3 HOW TO IMPROVE THE BIT-LEVEL SCALING LAWS OF GENERATIVE MODELS?

Given the observed differences in scaling results between language-style models and diffusion-style models, and the demonstrated advantages of language-style models, an important follow-up motivation is to enhance the bit-level scaling of language-style models. To this end, we conduct extensive experiments to investigate the impact of various recently studied advancements in quantization precision on the bit-level scaling laws of language-style models.

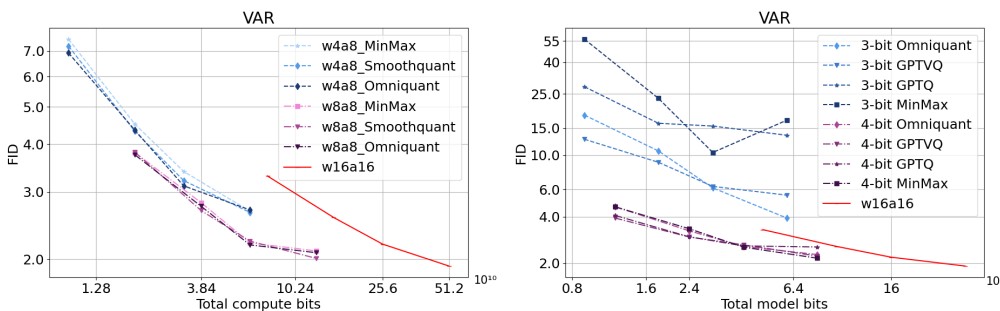

Figure 5: Comparison of bit-level scaling laws across various existing superior PTQ methods. Results show these methods exhibit only marginal improvements at W8A8 and W4A16, and performance significantly deteriorates at lower bit settings, suggesting that existing PTQ fail to substantially enhance bit-level scaling laws.

**No substantial scaling improvement with existing methods** We evaluated existing quantization methods and find that while they improve the scaling behavior of models at W3A16 and W4A8, the best bit-level scaling behavior is still observed at W8A8 and W4A16 settings. The main reason for this seems to be that the models retain sufficient precision at these precision levels, resulting in minimal degradation compared to full-precision models, and hence, there is not a significant enhancement in bit-level scaling, as shown in Figure 5. Lower bit precision often presents more promising scaling trends. Therefore, to improve the bit-level scaling laws, we aim to enhance the scaling behavior of

models specifically at W3A16 and W4A8. If you would like to further experimental results, please refer to Appendix A.

**Distillation for restoring the scaling at low bits**   To further enhance the model's scaling behavior at low bits, we apply knowledge distillation in Quantization-Aware Training (QAT), where the full-precision model serves as the teacher and its quantized variant as the student, learning the token-level probability distributions to more closely approximate the behavior of its full-precision counterpart. As shown in Figure 6, the model still exhibits optimal bit-level behavior at W4A6 and W8A8 precision. However, at lower bit levels, the scaling behavior closely approaches this optimal state, demonstrating that knowledge from the full-precision model, introduced through distillation, plays a crucial role in recovering scaling laws at lower bit precisions.

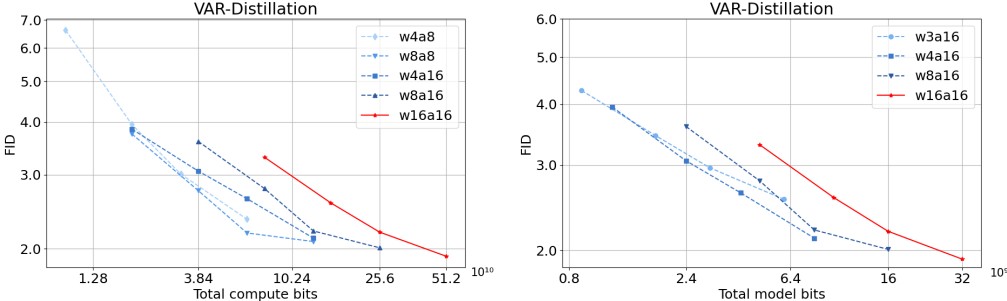

Figure 6: Visualization of bit-level scaling laws with distillation applied to QAT under VAR. With the use of distillation, the model's scaling behavior at lower bit precisions is restored to the level of higher bit precisions. Specifically, at W3A16, the scaling behavior nearly reaches that of W4A16. When both weights and activations are quantized, the scaling behavior at W4A8 closely approaches that observed at W8A8.

**Distillation with TopKLD for improving the scaling**   The choice of knowledge for distillation is crucial (Hinton, 2015; Zhu et al., 2023). (Agarwal et al., 2023) found that the mode-seeking behavior encouraged by the Reverse KL divergence (Gu et al., 2024) results in better fitting of "explicit knowledge" compared to the Forward KL divergence for instruction tuning tasks (Chung et al., 2024). However, (Zhao et al., 2022) reveals that the classic KD loss is a highly coupled form where non-target logits contain significant "implicit knowledge". **The Reverse KL divergence exacerbates the disregard for this knowledge, which is not preferable since the more confident the teacher model is in a training sample, the more severe the neglect of implicit knowledge becomes. Conversely, this implicit knowledge is more reliable and valuable.** For language-style models, top-k sampling is often employed to enhance generation quality (Ramesh et al., 2021; Tian et al., 2024). Therefore, we propose a customized approach that combines topk mode-seeking with others mode-covering techniques to balance the "implicit knowledge" and "explicit knowledge", called as TopKLD. In order to achieve this, we decompose the probability vector $P$ as follows, based on top-K sampling: $P = [M_s.M_c]$. Here, $M_s$ contains the probabilities of the top-K categories, which we aim to fit using mode-seeking techniques. $M_c$ includes the probabilities of the remaining categories, which we fit using mode-covering techniques. The proposed TopKLD can be represented by the following equation:

$$\text{TopKLD}(P_T||P_S) = \sum_{t=1,y'\in M_s}^{T} P_S(y'|x,y_{<t})\log\frac{P_S(y'|x,y_{<t})}{P_T(y'|x,y_{<t})} + \sum_{t=1,y'\in M_c}^{T} P_T(y'|x,y_{<t})\log\frac{P_T(y'|x,y_{<t})}{P_S(y'|x,y_{<t})} \tag{7}$$

Where, $P_T$ and $P_S$ denote the full-precision and quantized model, respectively. Figure 7 demonstrates the differences between Forward KLD, Reverse KLD, and TopKLD when a Gaussian distribution attempts to fit a Gaussian Mixture, along with their respective scaling behavior results under the W3A16 setting. It is evident that TopKLD effectively balances between "implicit knowledge" and "explicit knowledge", allowing for better utilization of the full-precision model's information. Figure 7 illustrates the bit-level scaling laws of the model using TopKLD under weight-only and weight-

activation quantization settings. It can be observed that the model exhibits improved scaling laws under W3A16 and W4A8 settings, further enhancing the scaling behavior of the model.

Additionally, floating-point (FP) quantization has become a promising alternative to integer quantization because of its capability to manage long-tail distributions and its greater flexibility (Kuzmin et al., 2022). We applied TopKLD distillation to FP quantization, demonstrating its applicability in floating-point settings and further improving the model's bit-level scaling behavior compared to integer quantization.

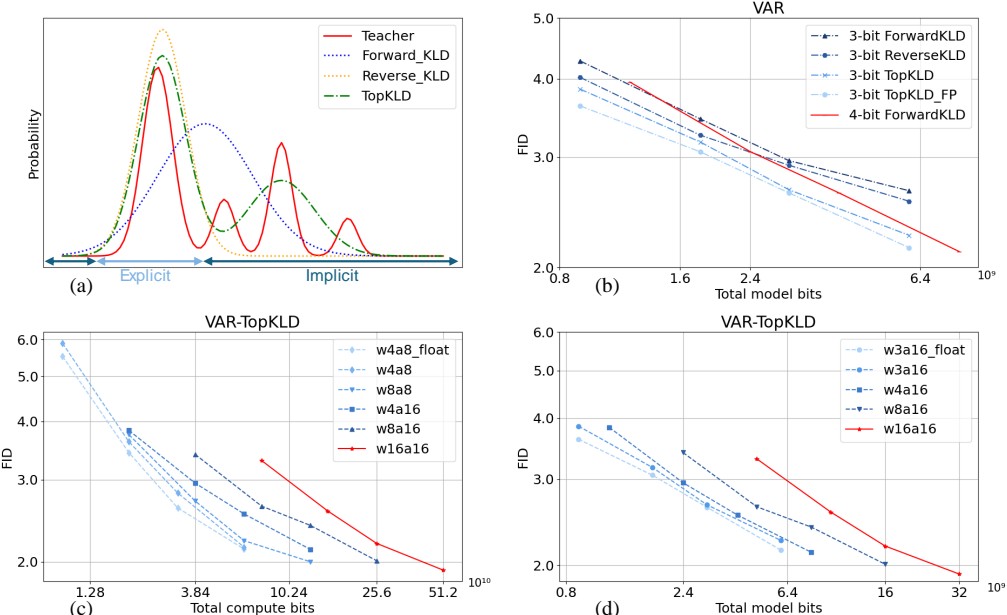

Figure 7: (a) Comparison of Reverse KL, Forward KL, and TopKLD when a Gaussian distribution attempts to fit a Gaussian mixture (Teacher). (b) Comparison of different KL divergences under W3A16, showing that TopKLD achieves the best performance, outperforming the optimal scaling behavior (4 bits) with ForwardKLD. (c-d) Visualization of bit-level scaling laws under TopKLD. Additionally, under floating-point datatype, TopKLD can further improve the scaling behavior.

## 4 RELATED WORK

**Large language model quantization.** As model scaling capabilities improve and the number of parameters increases, the most closely related work is on large language model (LLM) quantization for models with over a billion parameters. Compared to smaller models, larger models quantization poses some unique challenges, such as emergent outliers (Chee et al., 2024; Lin et al., 2024) and the need for optimized low-bit inference (Tseng et al., 2024; Dettmers et al., 2024). To address these issues, previous studies have proposed solutions like outlier processing and first- or second-order optimization. Methods such as SmoothQuant (Xiao et al., 2023), Outlier Suppression (Wei et al., 2022), and Outlier Suppression+ (Wei et al., 2023) focus on managing activation outliers, achieving promising results in W8A8 precision. GPTQ (Frantar et al., 2022) leverages second-order Hessian matrix optimization to adjust model weights, obtaining high accuracy in W4A16. Furthermore, techniques like OmniQuant (Shao et al., 2023) and QLLM (Liu et al., 2023) apply first-order gradient-based optimization for quantizing parameters, yielding strong results in models using 4-bit or higher precision settings.

**Visual model quantization** In visual model quantization, optimization has not advanced in line with the scaling capabilities of models. Instead, efforts have focused more on addressing the specific distribution characteristics of individual layers and the multi-timestep inference features of generative models. For example, FQ-ViT (Lin et al., 2021) introduces Powers-of-Two Scale and Log-Int-Softmax techniques to quantize LayerNorm and Softmax operations, enabling fully quantized models.

PTQ4ViT (Yuan et al., 2022) employs twin uniform quantization to manage unbalanced post-Softmax and post-GELU activation distributions, using a Hessian-guided metric for optimal quantization scales. PTQ4DM (Shang et al., 2023) and Q-diffusion (Li et al., 2023b) introduce tailored calibration samples designed to account for activation distribution variance across timesteps. HQ-DiT (Liu & Zhang, 2024) adaptively selects the optimal floating-point format based on the data distribution. PTQ4DiT (Wu et al., 2024) proposes a channel-wise salience balance between weight and activation, placing greater emphasis on enhancing complementarity across timesteps.

## 5 RECOMMENDATIONS & FUTURE WORK

A well-optimized bit-level scaling behavior could offer substantial benefits by enabling the fine-tuning of models to achieve higher efficiency and accuracy under constrained resource conditions. Our study underscores the significant potential of quantization in optimizing the bit-level scaling laws of visual generative models, particularly in language-style models. The results indicate that achieving optimal bit-level scaling behavior requires a synergistic interaction between model design and quantization algorithms. Our study is an essential step towards understanding how various models and quantization methods influence bit-level scaling behavior, and it also provides the following recommendations for future work.

**Exploration of Advanced Quantization Techniques** Our results demonstrate that while existing quantization methods provide a foundation for enhancing bit-level scaling, they fall short of fully optimizing the scaling behaviors at extremely low bit precisions (e.g.3 bits). This indicates that there is significant room for improvement, particularly in advancing the scaling performance of models operating under strict bit constraints. Future research should focus on developing advanced quantization techniques tailored to the unique characteristics of language-style and diffusion-style models. This could involve creating novel quantization strategies that specifically address the challenges associated with lower-bit scaling behavior.

**Optimization of Knowledge Distillation Techniques** Our experiments reveal that distillation is an excellent method for restoring bit-level scaling behavior. In this context, our proposed TopKLD method shows promise in balancing "implicit knowledge" and "explicit knowledge" to improve bit-level scaling. In the future, we will optimize this method further, potentially by integrating it with other knowledge distillation frameworks or exploring its effectiveness across different quantization settings and model architectures. The goal would be to develop a robust distillation strategy that consistently enhances bit-level scaling across a wide range of models.

**Investigating More Comprehensive Model Scaling Laws** Our work primarily focuses on diffusion-style and language-style models, particularly those that have clearly exhibited scaling laws, such as DIT and VAR. Expanding this research to encompass a wider array of visual generative models could offer a more comprehensive understanding of bit-level scaling laws. Furthermore, recent studies (Tschannen et al., 2023; Li et al., 2024a) have concentrated on continuous-valued tokens in sequence models. Exploring the applicability of our findings to these generative paradigms could provide valuable insights into the generalizability of these scaling laws.

## 6 CONCLUSION

Our study provides a comprehensive analysis of the distinct bit-level scaling behaviors in visual generative models, revealing key differences in their scaling performance. We found that the representation space reconstruction in language-style models offers a more stable foundation for scaling at low bit precision. Moreover, we introduced the TopKLD method, which enhances knowledge transfer from full-precision models by effectively balancing explicit and implicit knowledge, thereby improving the bit-level scaling performance of language-style models. Overall, our study offers new insights into the design of future quantization and visual model strategies that can optimize both memory efficiency and model accuracy.

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

# A DETAILS OF THE IMPACT OF EXISTING PTQ METHODS ON THE BIT-LEVEL SCALING LAWS OF VISUAL GENERATIVE MODELS.

These sections provide a comprehensive analysis of the effects of various superior PTQ methods on language-style vision generative models. We categorize past PTQ algorithms into three main types: first-order gradient optimization, second-order Hessian matrix optimization, and vector quantization. We select representative methods from each category to explore their influence on the bit-level scaling laws of visual generative models.

## A.1 DETAILED EXAMINATION OF FIRST-ORDER GRADIENT OPTIMIZATION IN POST-TRAINING QUANTIZATION

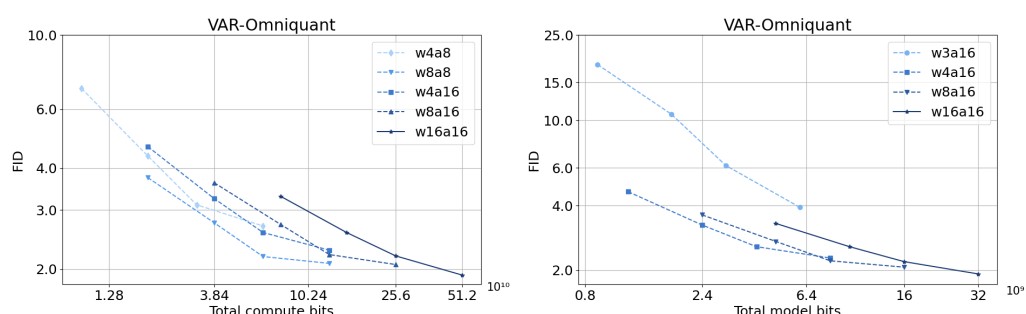

Figure 8: Bit-Level scaling laws based on first-order gradient optimization (Omniquant) in PTQ

PTQ methods based on first-order gradient optimization aim to minimize quantization error while adapting to both data and task-specific losses. In this framework, the objective is to optimize both quantization step sizes and zero points. This can be formulated as follows:

$$\arg\min_{S,Z} \mathbb{E}(T(W, X), T(Q(W), Q(X))) \tag{8}$$

where $S, Z$ represent the step sizes and zero points for activation and weight quantization, respectively. $\mathbb{E}(\cdot)$ measures the reconstruction error between the quantized and full-precision model, and $Q(\cdot)$ denotes a uniform quantizer. This formulation allows for the optimization of both step sizes and zero points across layers or blocks in models. Previous PTQ methods utilizing gradient optimization, such as AdaRound (Nagel et al., 2020) and BRECQ (Li et al., 2021), build upon this foundation. However, as the number of model parameters increases, it has been found that these methods cannot be effectively applied to models with billions of parameters due to the challenges in optimizing within the vast solution space. To address this issue, OmniQuant (Shao et al., 2023) introduces a novel optimization pipeline that minimizes block-wise quantization error, allowing additional quantization parameters to be optimized in a differentiable manner. We formulate the optimization goal as follows:

$$\arg\min_{\gamma,\beta,s} ||\mathcal{F}(\mathbf{W}, \mathbf{X}) - \mathcal{F}\big(Q_w(\mathbf{W}; \gamma, \beta), Q_a(\mathbf{X}, s, \delta)\big)|| \tag{9}$$

where $\mathcal{F}$ represents the mapping function for a transformer block in the model, $Q_w(\cdot)$ and $Q_a(\cdot)$ represent weight and activation quantizer, respectively, $\gamma$, $\beta$, $s$, $\delta$ are quantization parameters in learnable weight clipping and learnable equivalent transformation, which are defined as follows:

$$\mathbf{W_q} = \text{clamp}(\lfloor\frac{\mathbf{W}}{h}\rceil + z, 0, 2^N - 1), \text{where } h = \frac{\gamma\max(\mathbf{W}) - \beta\min(\mathbf{W})}{2^N - 1}, z = -\lfloor\frac{\beta\min(\mathbf{W})}{h}\rceil \tag{10}$$

$$\mathbf{Y} = \mathbf{X}\mathbf{W} + \mathbf{B} = [\underbrace{(\mathbf{X} - \delta) \oslash s}_{\tilde{\mathbf{X}}}] \cdot [\underbrace{s \odot \mathbf{W}}_{\tilde{\mathbf{W}}}] + [\underbrace{\mathbf{B} + \mathbf{W}}_{\tilde{\mathbf{B}}}] \tag{11}$$

Thus, we experiment with both weight-only and weight-activation quantization to assess their impact on bit-level scaling behavior. The results are as Figure 8 in the Appendix A.

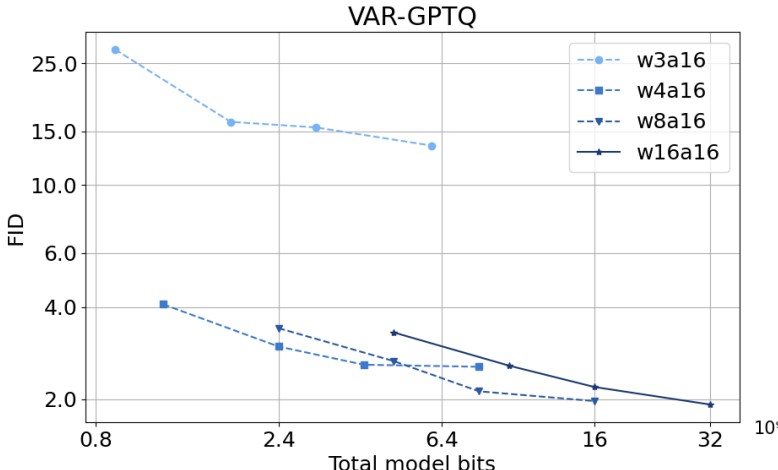

Figure 9: Bit-Level scaling laws based on second-order hessian matrix optimization (GPTQ) in PTQ

## A.2 DETAILED EXAMINATION OF SECOND-ORDER HESSIAN MATRIX OPTIMIZATION IN POST-TRAINING QUANTIZATION

To reduce the impact of quantization noise on model accuracy, Optimal Brain Quantization (OBQ) (Frantar & Alistarh, 2022) extends the Optimal Brain Surgeon (OBS) (LeCun et al., 1989) framework by incorporating second-order Hessian information to minimize quantization errors. The goal is to minimize the Hessian-weighted error introduced by quantizing weights $\mathbf{W}^{(\ell)}$:

$$E = \sum_q |E_q|_2^2; \qquad\qquad E_q = \frac{\mathbf{W}_{:,q} - \text{quant}(\mathbf{W}_{:,q})}{\left[\mathbf{H}^{-1}\right]_{qq}}. \qquad (12)$$

However, its cubic runtime makes OBQ impractical for large models with scaling characteristics. Specifically, for a $d_{\text{row}} \times d_{\text{col}}$ matrix $\mathbf{W}$, the runtime scales as $O(d_{\text{row}} \cdot d_{\text{col}}^3)$. To address these scalability issues, GPTQ (Frantar et al., 2022) improves on OBQ by quantizing all weights in a column simultaneously using a shared Hessian $\mathbf{H}^{(\ell)}$ across rows of wight $\mathbf{W}^{(\ell)}$. After quantizing a column $q$, the remaining columns $q' > q$ are updated using a Hessian-based rule $\delta$ to account for the quantization error in column $q$, which is given by:

$$\delta = -\frac{\mathbf{W}_{:,q} - \text{quant}(\mathbf{W}_{:,q})}{\left[\mathbf{H}^{-1}\right]_{qq}}\mathbf{H}_{:,(q+1):} \qquad (13)$$

To improve efficiency, GPTQ applies these updates in blocks of size $B$, reducing the amount of data transfer. The error $E_q$ in Equation 12 is accumulated while columns in block $B$ are processed and applied to the remaining columns afterward. Additionally, GPTQ uses a Cholesky decomposition of the inverse Hessian $\mathbf{H}^{-1}$, providing a more stable and efficient alternative to OBQ's Hessian updates. These modifications make GPTQ significantly faster and more scalable for large models while maintaining accuracy in low-bit quantization. We also tested its impact on the model's bit-level scaling laws under weight-only quantization, with the results shown in the Figure.9 of the Appendix A.

## A.3 DETAILED EXAMINATION OF VECTOR QUANTIZATION IN POST-TRAINING QUANTIZATION

Scalar quantization as presented in the previous section, is efficient but limited to equidistant spacing of representable points. A more flexible quantization approach is Vector quantization using higher-dimensional codebooks quantization. In vector quantization (VQ), each centroid in the codebook $C$ represents $d$ values, and each $d$-dimensional vector in $x$ is indexed into $C^d$, where $C^d$ is a codebook with $d$-dimensional entries (Gersho & Gray, 2012). Product quantization involves splitting a $D$-dimensional vector into multiple $d$-dimensional sub-vectors. GPTVQ (van Baalen et al., 2024)

extends GPTQ to vector quantization by quantizing $d$ columns at a time. Instead of rounding to the nearest centroid, GPTVQ selects the optimal centroid by minimizing:

$$j = \arg\min_m \left(\mathbf{x} - \mathbf{c}^{(m)}\right)^T \mathbf{H}^{(i)} \left(\mathbf{x} - \mathbf{c}^{(m)}\right). \tag{14}$$

Equation 14 is used for choosing the optimal assignment $j$ for data point $x^{(i)}$ and the corresponding inverse sub-Hessian $\mathbf{H}^{(i)}$. After quantizing $d$ columns, GPTVQ updates the remaining weights and applies the accumulated update in a single operation. To further reduce quantization error, multiple codebooks are used per layer, each assigned to a group of weights. the detailed scaling laws shown in the Figure.10 of the Appendix A.

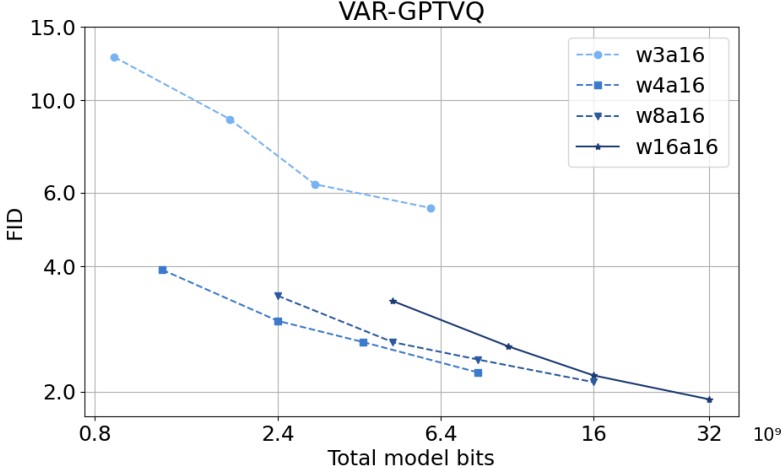

Figure 10: Bit-Level scaling laws based on Vector quantization (GPTVQ) in PTQ

## B COMPARISON OF ACTIVATION VALUE DISTRIBUTIONS VISUALIZATION

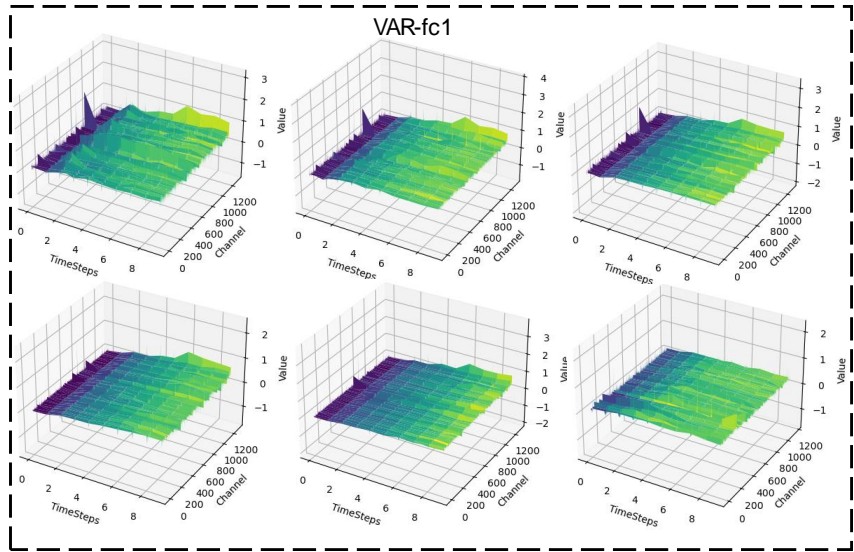

Figure 11: The visualization for the activation value distributions in the fc1 layers of VAR across the specified blocks (3rd, 6th, 9th, 13th, 16th, 19th).

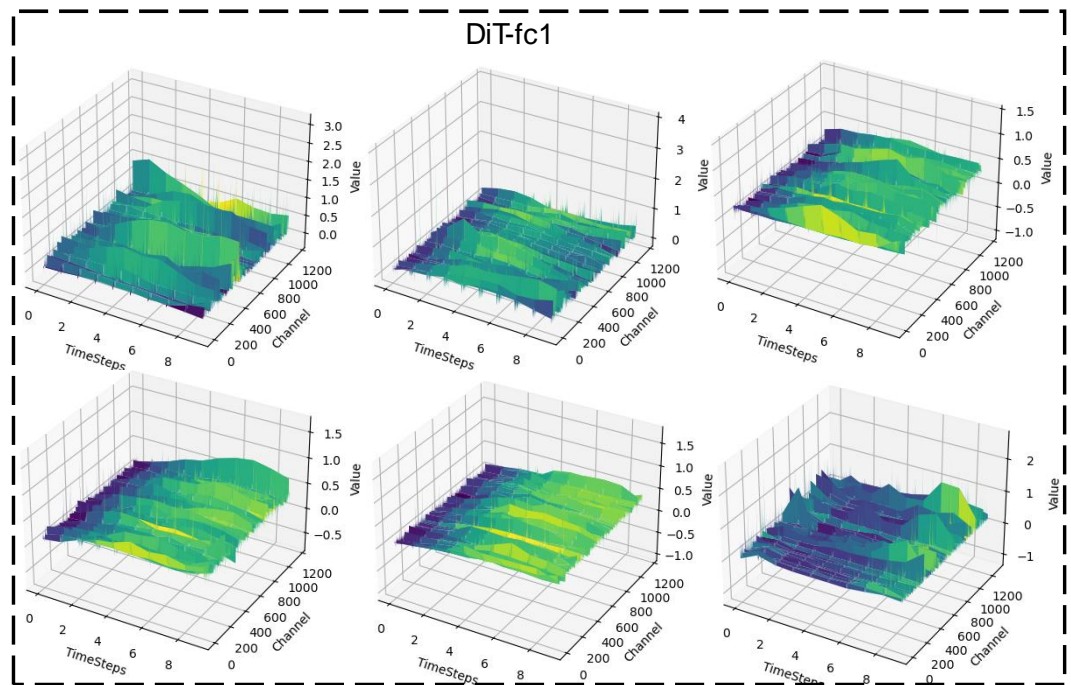

Figure 12: The visualization for the activation value distributions in the fc1 layers of DiT across the specified blocks (3rd, 6th, 9th, 13th, 16th, 19th).

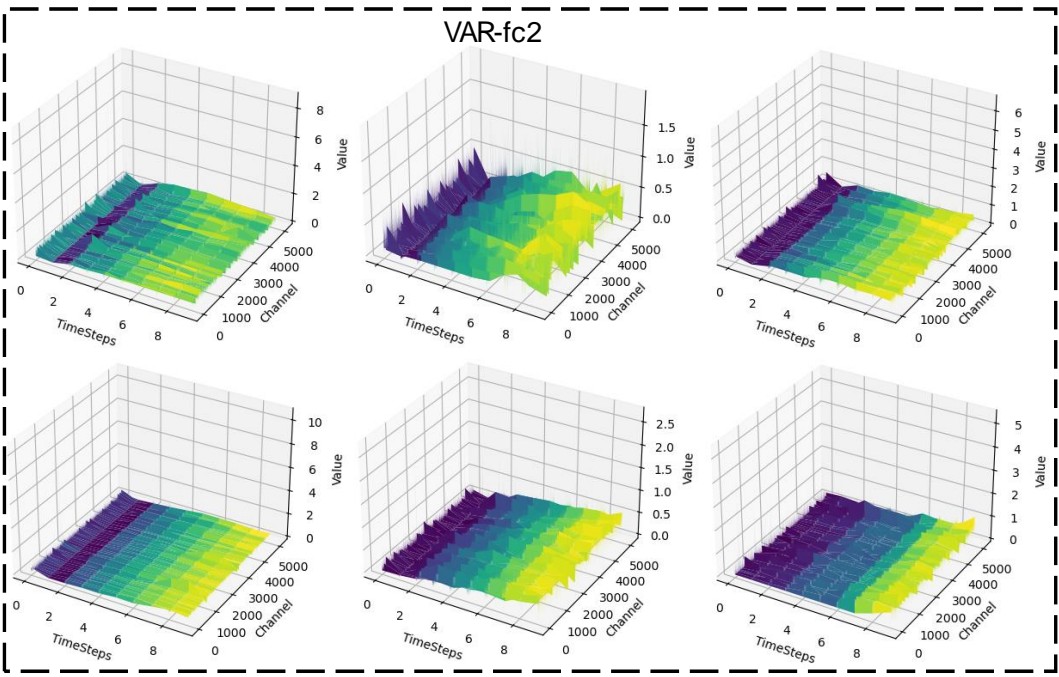

Figure 13: The visualization for the activation value distributions in the fc2 layers of VAR across the specified blocks (3rd, 6th, 7th, 9th, 11th, 19th).

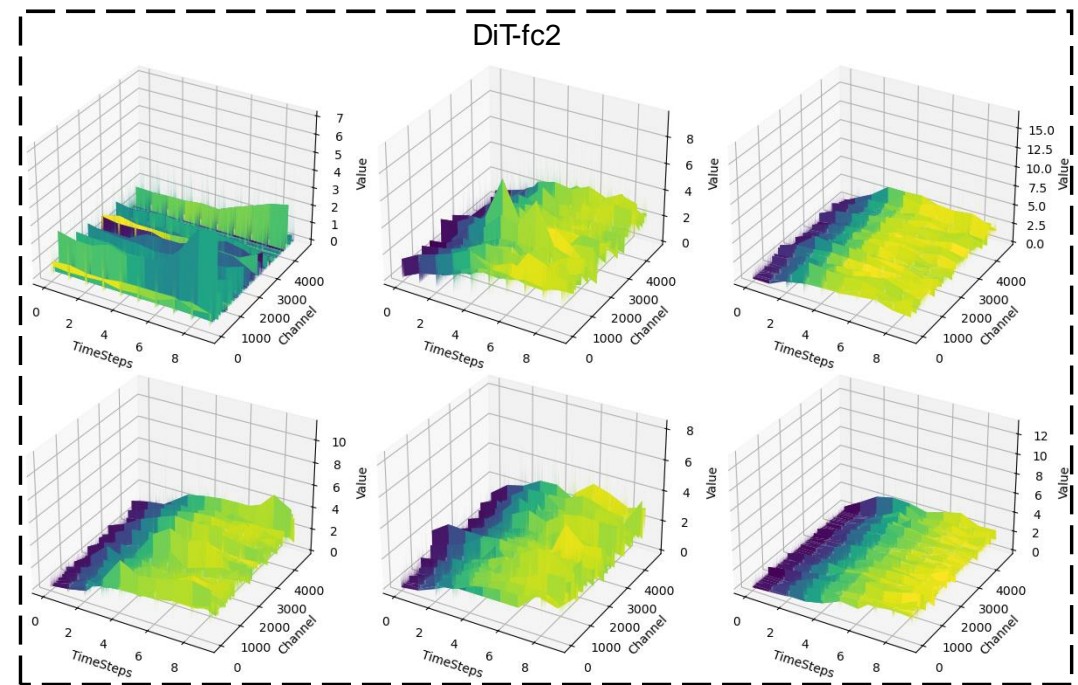

Figure 14: The visualization for the activation value distributions in the fc2 layers of DiT across the specified blocks (3rd, 6th, 7th, 9th, 11th, 19th).

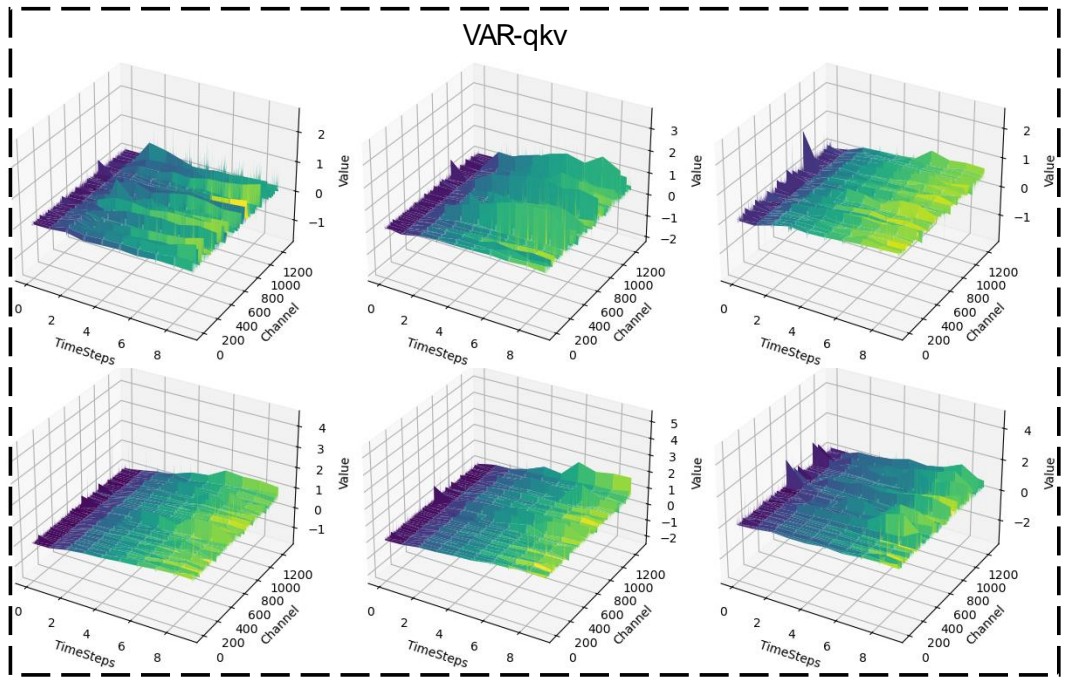

Figure 15: The visualization for the activation value distributions in the qkv layers of VAR across the specified blocks (3rd, 6th, 9th, 13th, 16th, 19th).

## C  SUPPLEMENT MATERIALS FOR REBUTTAL

### C.1  OVERVIEW OF VISUAL GENERATION MODELS

In the current landscape of vision generation models, there are two main development paths based on their generation mechanisms and representation spaces: language-style models and diffusion-style

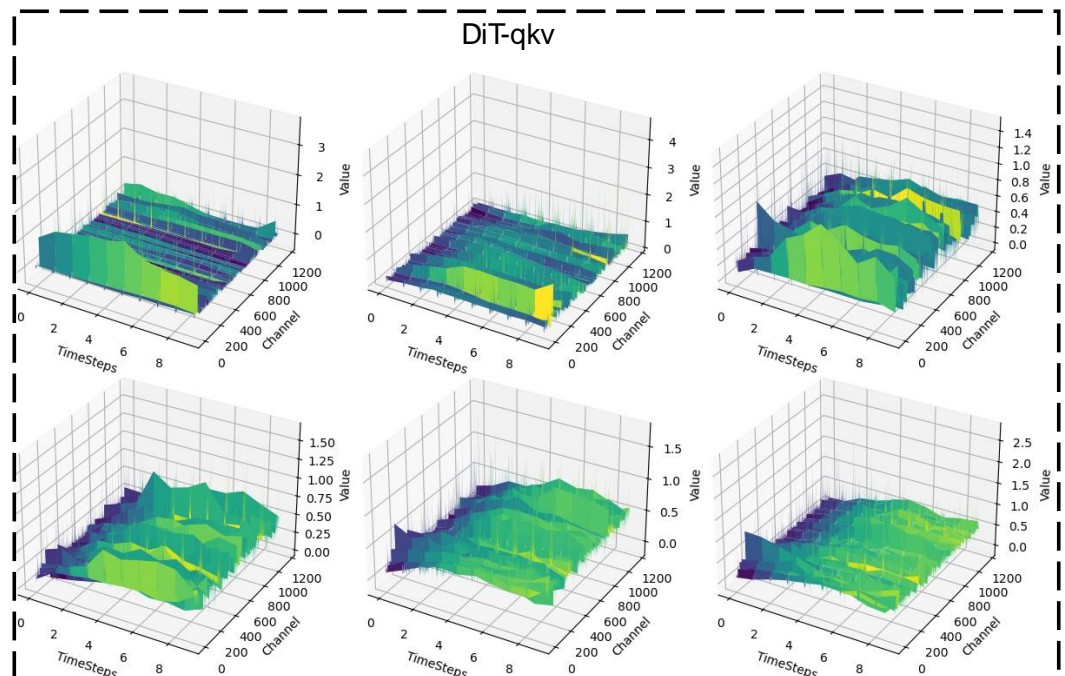

Figure 16: The visualization for the activation value distributions in the qkv layers of DiT across the specified blocks (3rd, 6th, 9th, 13th, 16th, 19th).

models, as well as models operating in discrete and continuous spaces. To systematically reveal the scaling laws, we present the following comparative table 1.

## C.2 EMPIRICAL VALIDATION THROUGH ADDITIONAL MODELS

To validate the generality of our findings in Section 3 regarding the experiments and conclusions on VAR and DIT, we conducted experiments using standard PTQ on two additional models that exhibit scaling laws: MAR (Li et al., 2024b) and LlamaGen (Sun et al., 2024)

MAR represents a continuous language-style model, aligning with the characteristics of DIT. Llama-Gen is a discrete language model, similar to VAR in terms of its discrete representation space. The results, as shown in the figure 17, reveal the following key observations:

MAR fails to exhibit superior bit-level scaling laws, consistent with our conclusion in Section 3.2. This can be attributed to its use of a continuous representation space, which is more sensitive to quantization effects.

LlamaGen demonstrates exceptional bit-level scaling laws. This aligns with our conclusion in Section 3.1, further confirming that discrete representation spaces provide significant advantages for bit-level scaling laws compared to continuous representation spaces.

These findings validate the broader applicability of our conclusions, reinforcing the importance of representation space choice in determining the scaling behavior of visual generation models.

Additionally, we also explored the effect of Top KLD on LlamaGen. Our experiments reveal that incorporating TopKLD significantly enhances the model's bit-level scaling laws, providing a more stable performance across various bit precisions,as shown in fig.18.

## C.3 COMPARISON OF TOP KLD WITH MAINSTREAM QUANTIZATION METHODS

To further highlight the advantages of Top KLD, we compared its performance with several mainstream quantization techniques, including Smoothquant (Xiao et al., 2023), Omniquant (Shao et al.,

Table 1: Scaling Laws and Characteristics of Vision Generation Models, where D-style and L-style represent Diffusion-style and Language-style vision generation models,respectively.

| Model Type | Discrete/Continuous | Model | #para | FID | IS | Dates | Scaling ability |
|---|---|---|---|---|---|---|---|
| D-style | Continuous | ADM (Dhariwal & Nichol, 2021) | 554M | 10.94 | 101 | 2021.07 | ✗ |
| | Continuous | CDM (Ho et al., 2022b) | - | 4.88 | 158.7 | 2021.12 | ✗ |
| | Continuous | LDM-8 (Rombach et al., 2022a) | 258M | 7.76 | 209.5 | 2022.04 | ✗ |
| | Continuous | LDM-4 (Rombach et al., 2022a) | 400M | 3.6 | 247.7 | | ✗ |
| | Continuous | DiT (Peebles & Xie, 2023) | 458M | 5.02 | 167.2 | 2023.03 | ✓ |
| | | | 675M | 2.27 | 278.2 | | |
| | | | 3B | 2.1 | 304.4 | | |
| | | | 7B | 2.28 | 316.2 | | |
| | Continuous | MDT (Gao et al., 2023) | 676M | 1.58 | 314.7 | 2024.02 | ✗ |
| | Continuous | DiMR (Liu et al., 2024a) | 505M | 1.7 | 289 | 2024.07 | ✗ |
| | Discrete | VQ-diffusion (Gu et al., 2022) | 370M | 11.89 | - | 2022.03 | ✗ |
| | Discrete | VQ-diffusion-V2 (Tang et al., 2022) | 370M | 7.65 | - | 2023.02 | ✗ |
| L-style | Discrete | MaskGIT (Chang et al., 2022) | 177M | 6.18 | 182.1 | 2022.02 | ✗ |
| | Discrete | RCG(cond.) (Li et al., 2023a) | 502M | 3.49 | 215.5 | 2023.12 | ✗ |
| | Discrete | MAGVIT-v2 (Yu et al., 2023b) | 307M | 1.78 | 319.4 | 2023.04 | ✗ |
| | Discrete | TiTok (Yu et al., 2024) | 287M | 1.97 | 281.8 | 2024.07 | ✗ |
| | Discrete | MaskBit (Weber et al., 2024) | 305M | 1.52 | 328.6 | 2024.09 | ✗ |
| | Discrete | VQVAE (Razavi et al., 2019a) | 13.5B | 31.11 | 45 | 2019.06 | ✗ |
| | Discrete | VQGAN (Esser et al., 2021a) | 1.4B | 5.2 | 175.1 | 2021.07 | ✗ |
| | Discrete | RQTran (Lee et al., 2022) | 3.8B | 3.8 | 323.7 | 2022.03 | ✗ |
| | Discrete | VITVQ (Yu et al., 2021) | 1.7B | 3.04 | 227.4 | 2022.07 | ✗ |
| | Discrete | VAR (Tian et al., 2024) | 310M | 3.3 | 274.4 | 2024.04 | ✓ |
| | | | 600M | 2.57 | 302.6 | | |
| | | | 1B | 2.09 | 312.9 | | |
| | | | 2B | 1.92 | 323.1 | | |
| | Discrete | LlamaGen (Sun et al., 2024) | 343M | 3.07 | 256.06 | 2024.07 | ✓ |
| | | | 775M | 2.62 | 244.1 | | |
| | | | 1.4B | 2.34 | 253.9 | | |
| | | | 3.1B | 2.18 | 263.3 | | |
| | Continuous | MAR (Li et al., 2024b) | 208M | 2.31 | 281.7 | 2024.07 | ✓ |
| | | | 479M | 1.78 | 296 | | |
| | | | 943M | 1.55 | 303.7 | | |

2023), GPTQ (Frantar et al., 2022), GPTVQ (van Baalen et al., 2024). Our analysis shows that Top KLD consistently achieves the SOTA results across various bit settings.

Table 2: Comparison of Top KLD with Mainstream Quantization Methods under weight-only quantization

| #bit | Method | d16 | d20 | d24 | d30 |
|---|---|---|---|---|---|
| W16A16 | FP16 | 3.3 | 2.57 | 2.19 | 1.92 |
| W8A16 | GPTQ | 3.41 | 2.66 | 2.12 | 1.97 |
| | GPTVQ | 3.40 | 2.637 | 2.398 | 2.11 |
| | OmniQ | 3.62 | 2.72 | 2.2098 | 2.0636 |
| | Forward-KLD | 3.41 | 2.636 | 2.40 | 2.05 |
| | Reverse-KLD | 3.41 | 2.636 | 2.41 | 2.04 |
| | TopKLD | 3.40 | 2.634 | 2.394 | 2.01 |
| W4A16 | GPTQ | 4.64 | 3.247 | 2.572 | 2.277 |
| | GPTVQ | 3.92 | 2.96 | 2.634 | 2.226 |
| | OmniQ | 4.08 | 3.17 | 2.56 | 2.55 |
| | Forward-KLD | 3.95 | 3.06 | 2.63 | 2.21 |
| | Reverse-KLD | 3.89 | 3.05 | 2.59 | 2.18 |
| | TopKLD | 3.82 | 2.95 | 2.53 | 2.12 |
| W3A16 | GPTQ | 27.75 | 16.11 | 15.45 | 13.48 |
| | GPTVQ | 12.69 | 9.01 | 6.29 | 5.52 |
| | OmniQ | 18.18 | 10.67 | 6.15 | 3.93 |
| | Forward-KLD | 4.27 | 3.45 | 2.96 | 2.55 |
| | Reverse-KLD | 4.02 | 3.25 | 2.91 | 2.55 |
| | TopKLD | 3.85 | 3.17 | 2.66 | 2.25 |

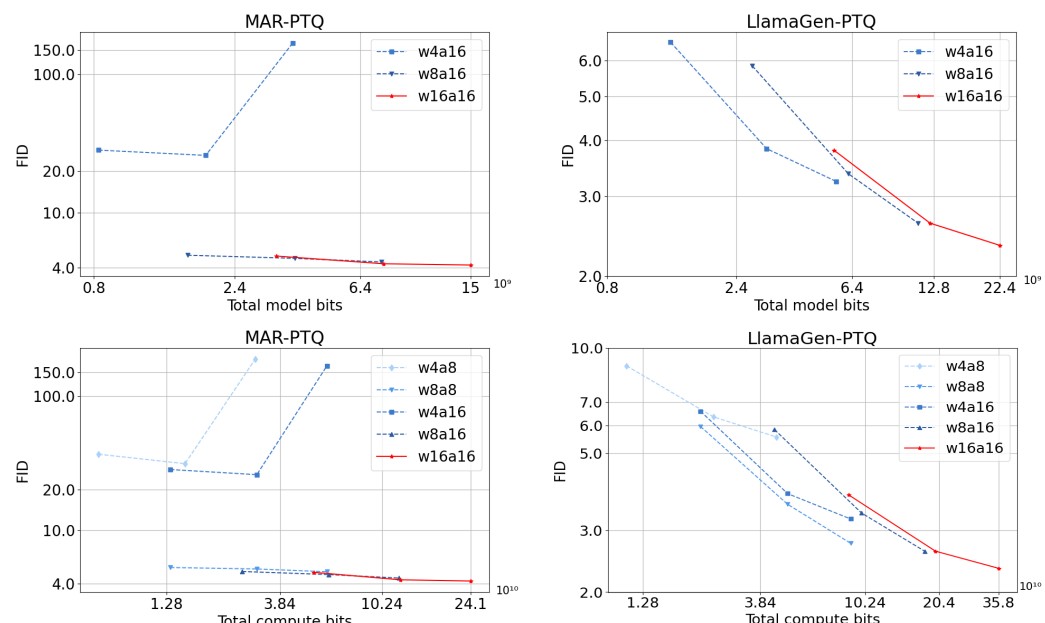

Figure 17: Investigation of bit-level scaling laws for MAR (left) and LlamaGen (right) models using standard PTQ. right: Quantited LlamaGen exhibits better bit-level scaling laws than full-precision LlamaGen.

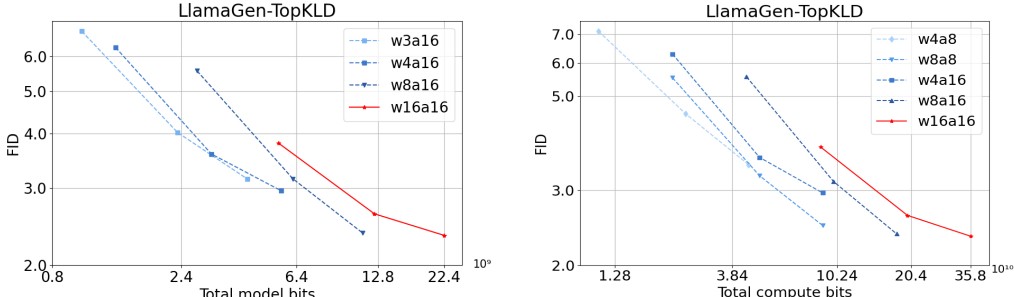

Figure 18: TopKLD provides a stable enhancement to the bit-level scaling ability of LlamaGen, particularly in the low-bit settings of W3A16 and W4A8.

## C.4    THE ABLATION OF TOPKLD

To further investigate the effectiveness of Top KLD, we conducted an ablation study to assess the impact of different components of the method on model performance,as shown in table 4 and 5

To understand the impact of Top-K sampling on the model's bit-level scaling, we conducted an ablation study with different values of K. The results shown in the table below reveal the following: (1) While the choice of K can influence the final generation quality to some extent, it does not affect the overall trend of the bit-level scaling laws. (2) The best performance occurs when the value of K matches the Top-K sampling used by the model during image generation.

Table 3: Comparison of Top KLD with Mainstream Quantization Methods under weight-activation quantization

| #bits | Method | d16 | d20 | d24 | d30 |
|---|---|---|---|---|---|
| W16A16 | FP | 3.3 | 2.57 | 2.19 | 1.92 |
| W8A8 | SmoothQ | 3.81 | 2.68 | 2.23 | 2.01 |
| | OmniQ | 3.75 | 2.75 | 2.18 | 2.08 |
| | ForwardKLD | 3.8 | 2.72 | 2.16 | 2.10 |
| | TopKLD | 2.75 | 2.7 | 2.18 | 1.98 |
| W4A8 | SmoothQ | 7.21 | 4.32 | 3.21 | 2.65 |
| | OmniQ | 6.92 | 4.35 | 3.11 | 2.69 |
| | ForwardKLD | 6.62 | 3.95 | 3.01 | 2.35 |
| | TopKLD | 5.89 | 3.62 | 2.81 | 2.15 |

Table 4: Ablation of TopKLD

| #bits | Method | d16 | d20 | d24 | d30 |
|---|---|---|---|---|---|
| W16A16 | FP | 3.3 | 2.57 | 2.19 | 1.92 |
| W3A16 | TopKLD (K = 400) | 3.95 | 3.21 | 2.77 | 2.29 |
| | TopKLD (K = 500) | 3.91 | 3.24 | 2.71 | 2.24 |
| | TopKLD (K = 600) | 3.85 | 3.17 | 2.66 | 2.25 |
| | TopKLD (K = 700) | 3.92 | 3.19 | 2.72 | 2.25 |
| | TopKLD (K = 800) | 3.96 | 3.19 | 2.73 | 2.29 |

Table 5: Ablation of TopKLD

| #Bits | Method | d16 | d20 | d24 | d30 |
|---|---|---|---|---|---|
| W16A16 | FP16 | 3.3 | 2.57 | 2.19 | 1.92 |
| W8A16 | MSE | 3.55 | 2.71 | 2.35 | 2.05 |
| | JS Divergence | 3.50 | 2.69 | 2.22 | 2.05 |
| | Forward-KLD | 3.41 | 2.636 | 2.40 | 2.05 |
| | Reverse-KLD | 3.41 | 2.636 | 2.41 | 2.04 |
| | TopKLD | 3.40 | 2.634 | 2.394 | 2.01 |
| W4A16 | MSE | 3.97 | 3.12 | 2.69 | 2.25 |
| | JS Divergence | 3.92 | 3.01 | 2.65 | 2.23 |
| | Forward-KLD | 3.95 | 3.06 | 2.63 | 2.21 |
| | Reverse-KLD | 3.89 | 3.05 | 2.59 | 2.18 |
| | TopKLD | 3.82 | 2.95 | 2.53 | 2.12 |
| W3A16 | MSE | 4.56 | 3.89 | 3.54 | 3.01 |
| | JS Divergence | 4.45 | 3.72 | 3.25 | 2.51 |
| | Forward-KLD | 4.27 | 3.45 | 2.96 | 2.55 |
| | Reverse-KLD | 4.02 | 3.25 | 2.91 | 2.55 |
| | TopKLD | 3.85 | 3.17 | 2.66 | 2.25 |

