# OpenReview forum: "Dissecting Bit-Level Scaling Laws in Quantizing Vision Generative Models"
_ICLR.cc/2025/Conference — Submitted to ICLR 2025_

### Official Review · Reviewer_7XYk · 2024-10-30

**Soundness:** 3
**Presentation:** 3
**Contribution:** 3
**Rating:** 5
**Confidence:** 3

**Summary:**

This paper explores scaling laws for model quantification. Besides, TopKLD is introduced to lift the decoder-only model's bit-level scaling performance.

**Strengths:**

1. This paper conducted many experiments based on VAR and DIT to explore the scaling law at the bit level.
2. The language-based model enjoys a better bit-level scaling law. The conclusion is interesting.
3. TopKLD seems effective in various quantitative aspects of VAR.

**Weaknesses:**

1. The paper is more like an experimental report than a research paper. I think the comparison between VAR and DIT is too lengthy and the TopKLD is short.
2. The model size of VAR is small. Is the necessity of quantifying small models sufficient?
3. Can you provide a direct visualization result that clearly shows the bit-level scaling law?

**Questions:**

See weakness.

---

> ### Author Response · Authors · 2024-11-20
>
> # **Rebuttal Revision Paper Modifications**
>
> We greatly appreciate your valuable review comments. We have revised the paper according to your suggestions and submitted the rebuttal version. **For detailed modifications, please refer to the rebuttal version PDF and appendix C: Supplementary materials for rebuttal.** Below, we address your identified weaknesses and questions, hoping to resolve your concerns and improve our score.
>
> # **Table 1**
>
> | Model   Type | Discrete/Continuous | model | #para | FID | IS | dates | Scaling ability |
> |:---:|:---:|:---:|:---:|:---:|:---:|:---:|:---:|
> | Diffusion-style | continuous | ADM[1] | 554M | 10.94 | 101 | 2021.07 | No |
> | Diffusion-style | continuous | CDM[2] | - | 4.88 | 158.7 | 2021.12 | No |
> | Diffusion-style | continuous | LDM-8[3] | 258M | 7.76 | 209.5 | 2022.04 | No |
> | Diffusion-style | continuous | LDM-4 | 400M | 3.6 | 247.7 |  | No |
> | Diffusion-style | continuous | DiT[4] | 458M | 5.02 | 167.2 | 2023.03 | Yes |
> | Diffusion-style |  |  | 675M | 2.27 | 278.2 |  |  |
> | Diffusion-style |  |  | 3B | 2.1 | 304.4 |  |  |
> | Diffusion-style |  |  | 7B | 2.28 | 316.2 |  |  |
> | Diffusion-style | continuous | MDTv[5] | 676M | 1.58 | 314.7 | 2024.02 | No |
> | Diffusion-style | continuous | DiMR[6] | 505M | 1.7 | 289 | 2024.07 | No |
> | Diffusion-style | Discrete | VQ-diffusion[7] | 370M | 11.89 | - | 2022.03 | No |
> | Diffusion-style | Discrete | VQ-diffusion-V2[8] | 370M | 7.65 | - | 2023.02 |  |
> | Language-style | Discrete | MaskGIT[9] | 177M | 6.18 | 182.1 | 2022.02 | No |
> | Language-style | Discrete | RCG(cond.)[10] | 502M | 3.49 | 215.5 | 2023.12 | No |
> | Language-style | Discrete | MAGVIT-v2[11] | 307M | 1.78 | 319.4 | 2023.04 | No |
> | Language-style | Discrete | TiTok[12] | 287M | 1.97 | 281.8 | 2024.07 | No |
> | Language-style | Discrete | MaskBit[13] | 305M | 1.52 | 328.6 | 2024.09 | No |
> | Language-style | Discrete | VQVAE[14] | 13.5B | 31.11 | 45 | 2019.06 | No |
> | Language-style | Discrete | VQGAN[15] | 1.4B | 5.2 | 175.1 | 2021.07 | No |
> | Language-style | Discrete | RQTran[16] | 3.8B | 3.8 | 323.7 | 2022.03 | No |
> | Language-style | Discrete | VITVQ[17] | 1.7B | 3.04 | 227.4 | 2022.07 | No |
> | Language-style | Discrete | VAR[18] | 310M | 3.3 | 274.4 | 2024.04 | yes |
> | Language-style |  |  | 600M | 2.57 | 302.6 |  |  |
> | Language-style |  |  | 1B | 2.09 | 312.9 |  |  |
> | Language-style |  |  | 2B | 1.92 | 323.1 |  |  |
> | Language-style | Discrete | LlamaGen[19] | 343M | 3.07 | 256.06 | 2024.07 | yes |
> | Language-style |  |  | 775M | 2.62 | 244.1 |  |  |
> | Language-style |  |  | 1.4B | 2.34 | 253.9 |  |  |
> | Language-style |  |  | 3.1B | 2.18 | 263.3 |  |  |
> | Language-style | continuous | MAR[20] | 208M | 2.31 | 281.7 | 2024.07 | yes |
> | Language-style |  |  | 479M | 1.78 | 296 |  |  |
> | Language-style |  |  | 943M | 1.55 | 303.7 |  |  |

---

> ### Author Response · Authors · 2024-11-20
>
> # **Weakness 1**
>
> We greatly appreciate your valuable review comments. We apologize for any confusion caused by the phrasing in the paper and hope our response can clarify your concerns regarding the statement: "The paper is more like an experimental report than a research paper." Furthermore, TopKLD is just one part of our research. **The goal of this paper is not merely to propose an improvement to existing methods, but rather to conduct an in-depth study of the bit-level scaling laws in vision generative models, addressing the "What," "Why," and "How" from the perspective of extensive experimental design.**
>
> **Exploration of bit-level scaling laws must be based on the internal patterns derived from a large number of experiments.** These patterns can guide future research, which is why we conducted numerous experiments, as it is essential for uncovering these insights.
>
> The analysis of VAR and DiT represents research into two mainstream development directions **in the vision generative model field. As shown in Table 1 above, there has been ongoing debate regarding the use of discrete versus continuous representation spaces (e.g., [17,18,19,20]).** Both approaches have shown strong performance in terms of scaling laws. **This work, however, takes a different perspective by investigating the impact of these representation spaces on the scaling laws in quantized models.** We find that, despite achieving comparable performance at full precision, discrete autoregressive models consistently outperform continuous models across various quantization settings. **To validate the effectiveness and broad applicability of our conclusions for you, we conducted the same experiments on other models, as detailed in Appendix C. This indicates that our work provides general guidance for subsequent model design and applications in specific deployment scenarios (e.g., mobile devices, edge computing).**
>
> Secondly, while low-bit precision representation often focuses on trading performance for efficiency, **this work demonstrates that by optimizing either the model or quantization algorithm, models can achieve superior bit-level scaling laws.** This outstanding characteristic enables the use of lower bit precision to increase model parameters, ultimately **enhancing generative capability without sacrificing efficiency — is a key feature that we hope researchers will pay particular attention to.**
>
> As such, you can see the tremendous potential of low-bit precision in the context of bit-level scaling laws. However, **existing methods fail to further improve the bit-level scaling laws of models.** To address this, we introduced **TopKLD, which enhances the bit-level scaling behaviors of language-style models by one level.**
>
> **Our study is an essential step toward understanding how various models and quantization methods influence bit-level scaling behavior, and it also provides the following recommendations for future work:** We hope the reviewer will take into account the contributions of this work to model design and the application of quantization algorithms. Thank you again!

---

> ### Author Response · Authors · 2024-11-20
>
> # **Weakness 2**
>
> Thank you very much for your suggestion.  **To address your concerns regarding the model, we conducted the same experiments on other models, as detailed in Appendix C.** It can be observed that due to the influence of the continuous representation space, MAR, despite exhibiting excellent scaling laws, do not demonstrate superior bit-level scaling laws,similar to DiT. In contrast, LLaMaGen, which shares the discrete representation space with VAR, exhibits outstanding bit-level scaling laws.
>
> Additionally, we provide an explanation of the impact of model size on bit-level scaling laws based on their underlying principles. Firstly, **Our research focuses on analyzing the differences in scaling trends between models that already exhibit superior scaling laws, rather than being influenced by specific model sizes. To ensure clarity, we have aligned the initial total bits in Figure 1 of paper, providing you with a clearer understanding.** To assess whether a model exhibits strong bit-level scaling laws, one must compare the internal change trends of the model (e.g., comparing 8-bit VAR vs. 16-bit VAR). As shown in Figure 1 in paper, we can observe that, regardless of the quantization method, when the full-precision VAR is quantized to lower bit precision, the overall scaling law of the model shifts towards the lower-left corner. However, DiT does not exhibit this behavior. This outstanding characteristic displayed by the discrete model enables us to increase the model parameters through quantization under limited resources, leading to better generative performance, which is not possible for continuous models. This is precisely what bit-level scaling laws aim to demonstrate.
>
> More importantly, **our work shows that quantization is no longer just about reducing model size.** By optimizing both model design and quantization techniques to achieve superior bit-level scaling laws, **we can obtain better generative performance under the same resource constraints. This outstanding feature is something that researchers should pay more attention to.**
>
>
>
>
> # **Weakness 3**
>
> We apologize for any confusion caused by the phrasing in the paper. We hope that the explanation of Figure 1 in the main text helps clarify the concept of bit-level scaling laws and this interesting phenomenon.. **As shown in the figure 1, quantized VAR, after being quantized to lower bit precision, demonstrates a shift in its scaling law curve towards the lower-left region, which exhibits its superior bit-level scaling laws.** By leveraging this outstanding feature, we can increase the model parameters under limited resources (e.g., in specific deployment scenarios such as mobile devices or edge computing) while maintaining efficiency, ultimately improving generative capabilities. In contrast, for continuous diffusion-style models, regardless of the quantization method used, the quantized model shows "almost" no improvement compared to full precision. Bit-level scaling laws serve as a strong predictor of model performance.
>
> This paper indicates that achieving optimal bit-level scaling behavior requires a synergistic interaction between model design and quantization algorithms. **Our study is an essential step towards understanding how various models and quantization methods influence bit-level scaling behavior, and it also provides valuable recommendations for future work.**

---

> ### Author Response · Authors · 2024-11-20
>
> # **Reference**
> [1]Dhariwal P, Nichol A. Diffusion models beat gans on image synthesis[J]. Advances in neural information processing systems, 2021, 34: 8780-8794.
>
> [2] Ho J, Saharia C, Chan W, et al. Cascaded diffusion models for high fidelity image generation[J]. Journal of Machine Learning Research, 2022, 23(47): 1-33.
>
> [3] Rombach R, Blattmann A, Lorenz D, et al. High-resolution image synthesis with latent diffusion models[C]//Proceedings of the IEEE/CVF conference on computer vision and pattern recognition. 2022: 10684-10695.
>
> [4] Peebles W, Xie S. Scalable diffusion models with transformers[C]//Proceedings of the IEEE/CVF International Conference on Computer Vision. 2023: 4195-4205.
>
> [5] Gao S, Zhou P, Cheng M M, et al. Masked diffusion transformer is a strong image synthesizer[C]//Proceedings of the IEEE/CVF International Conference on Computer Vision. 2023: 23164-23173.
>
> [6] Liu Q, Zeng Z, He J, et al. Alleviating Distortion in Image Generation via Multi-Resolution Diffusion Models[J]. arXiv preprint arXiv:2406.09416, 2024.
>
> [7] Gu S, Chen D, Bao J, et al. Vector quantized diffusion model for text-to-image synthesis[C]//Proceedings of the IEEE/CVF conference on computer vision and pattern recognition. 2022: 10696-10706.
>
> [8] Tang Z, Gu S, Bao J, et al. Improved vector quantized diffusion models[J]. arXiv preprint arXiv:2205.16007, 2022.
>
> [9] Chang H, Zhang H, Jiang L, et al. Maskgit: Masked generative image transformer[C]//Proceedings of the IEEE/CVF Conference on Computer Vision and Pattern Recognition. 2022: 11315-11325.
>
> [10] Li T, Katabi D, He K. Self-conditioned image generation via generating representations[J]. arXiv preprint arXiv:2312.03701, 2023.
>
> [11] Yu L, Lezama J, Gundavarapu N B, et al. Language Model Beats Diffusion--Tokenizer is Key to Visual Generation[J]. arXiv preprint arXiv:2310.05737, 2023.
>
> [12] Yu Q, Weber M, Deng X, et al. An Image is Worth 32 Tokens for Reconstruction and Generation[J]. arXiv preprint arXiv:2406.07550, 2024.
>
> [13] Weber M, Yu L, Yu Q, et al. Maskbit: Embedding-free image generation via bit tokens[J]. arXiv preprint arXiv:2409.16211, 2024.
>
> [14] Razavi A, Van den Oord A, Vinyals O. Generating diverse high-fidelity images with vq-vae-2[J]. Advances in neural information processing systems, 2019, 32.
>
> [15] Esser P, Rombach R, Ommer B. Taming transformers for high-resolution image synthesis[C]//Proceedings of the IEEE/CVF conference on computer vision and pattern recognition. 2021: 12873-12883.
>
> [16] Lee D, Kim C, Kim S, et al. Autoregressive image generation using residual quantization[C]//Proceedings of the IEEE/CVF Conference on Computer Vision and Pattern Recognition. 2022: 11523-11532.
>
> [17] Yu J, Li X, Koh J Y, et al. Vector-quantized image modeling with improved vqgan[J]. arXiv preprint arXiv:2110.04627, 2021.
>
> [18] Tian K, Jiang Y, Yuan Z, et al. Visual autoregressive modeling: Scalable image generation via next-scale prediction[J]. arXiv preprint arXiv:2404.02905, 2024.
>
> [19] Sun P, Jiang Y, Chen S, et al. Autoregressive Model Beats Diffusion: Llama for Scalable Image Generation[J]. arXiv preprint arXiv:2406.06525, 2024.
>
> [20] Chung Y A, Tang H, Glass J. Vector-quantized autoregressive predictive coding[J]. arXiv preprint arXiv:2005.08392, 2020.

---

> ### Author Response · Authors · 2024-11-22
>
> Dear reviewer:
>
> Thanks you for your great efforts in reviewing out paper and providing constructive suggestions/comments. **To address the weaknesses you raised, we have conducted extensive experiments in appendix C and figure 1 of main paper to alleviate concerns regarding the size of the VAR model.** Additionally, this work focuses on investigating the impact of whether the representation space in vision generation models is continuous or discrete. Furthermore, we propose strategies to optimize bit-level scaling laws under various quantization scenarios. **Our exploration of model design and quantization methods provides significant insights for guiding future applications in specific deployment scenarios, such as mobile devices and edge computing.** If our rebuttal does not address your concerns, you are warmly wecomed to raise questions. If our responses have addressed your concerns, we sincerely request that you consider raising our score.
>
> Best Wishes!
>
> Authors

---

> ### Author Response · Authors · 2024-11-27
>
> Dear reviewer:
>
> Thanks you for your great efforts in reviewing out paper and providing constructive suggestions/comments. **To address the weaknesses you raised, we have conducted extensive experiments in appendix C and figure 1 of main paper to alleviate concerns.** If our rebuttal does not address your concerns, you are warmly wecomed to raise questions. If our responses have addressed your concerns, we sincerely request that you consider raising our score.
>
> Best Wishes!
>
> Authors

---

> ### Author Response · Authors · 2024-11-29
>
> Dear reviewer:
>
> Thanks you for your great efforts in reviewing out paper and providing constructive suggestions/comments. **To address the weaknesses you raised, we have conducted extensive experiments in appendix C and figure 1 of main paper to alleviate concerns.** If our rebuttal does not address your concerns, you are warmly wecomed to raise questions. If our responses have addressed your concerns, we sincerely request that you consider raising our score.
>
> Best Wishes!
>
> Authors

---

### Official Review · Reviewer_gPYq · 2024-11-03

**Soundness:** 3
**Presentation:** 3
**Contribution:** 2
**Rating:** 5
**Confidence:** 4

**Summary:**

This paper investigates the impact of quantization on the performance of image generation models. By comprehensive experiments in many aspects, such as “model bits (MT), compute bits (CT)”, “post-training quantization (PTQ), quantization-aware training (QAT)”, “diffusion model (DiT), auto-regressive model (VAR)”, the authors observe that image generation models have  bit-level scaling laws. And they further discover that VAR is more robust to quantization than DiT due to its discrete representation space. Finally, they propose a knowledge distillation based quantization method, called TopKLD, to improve the bit-level scaling laws of VAR.

**Strengths:**

This paper demonstrates the bit-level scaling laws of image generative models through comprehensive experiments in terms of model bits and compute bits. By analysis of the reconstruction error of middle representations in VAR and DiT, the paper draws the conclusion that VAR is more robust to quantization and could generalize to other discrete auto-regressive models. And further, the paper proposes TopKLD, a quantization-aware training process, to improve scaling behavior of VAR at low bits region.

**Weaknesses:**

Bit-level scaling laws and the robustness of discrete auto-regressive models seem to be intuitive and straightforward, therefore the main contribution of this paper is the proposed quantization method, TopKLD. As a knowledge distillation based quantization-aware training method, the comparison and ablation studies are not enough.

**Questions:**

1. TopKLD should be compared to more distillation loss functions besides of forward and reverse KL Divergence, such as Logits MSE, JS Divergence and so on.
2. How does the parameter of “top-K sampling” affect the scaling behavior should be studied.
3. The “Figure 5” in line 427 should be “Figure 7(a)”

---

> ### Author Response · Authors · 2024-11-21
>
> # **Rebuttal Revision Paper Modifications**
>
> We greatly appreciate your valuable review comments. We have revised the paper according to your suggestions and submitted the rebuttal version. **For detailed modifications, please refer to the rebuttal version PDF and appendix C: Supplementary materials for rebuttal.** Below, we address your identified weaknesses and questions, hoping to resolve your concerns and improve our score.
>
> # **Table 1**
>
> | Model   Type | Discrete/Continuous | model | #para | FID | IS | dates | Scaling ability |
> |:---:|:---:|:---:|:---:|:---:|:---:|:---:|:---:|
> | Diffusion-style | continuous | ADM[1] | 554M | 10.94 | 101 | 2021.07 | No |
> | Diffusion-style | continuous | CDM[2] | - | 4.88 | 158.7 | 2021.12 | No |
> | Diffusion-style | continuous | LDM-8[3] | 258M | 7.76 | 209.5 | 2022.04 | No |
> | Diffusion-style | continuous | LDM-4 | 400M | 3.6 | 247.7 |  | No |
> | Diffusion-style | continuous | DiT[4] | 458M | 5.02 | 167.2 | 2023.03 | Yes |
> | Diffusion-style |  |  | 675M | 2.27 | 278.2 |  |  |
> | Diffusion-style |  |  | 3B | 2.1 | 304.4 |  |  |
> | Diffusion-style |  |  | 7B | 2.28 | 316.2 |  |  |
> | Diffusion-style | continuous | MDTv[5] | 676M | 1.58 | 314.7 | 2024.02 | No |
> | Diffusion-style | continuous | DiMR[6] | 505M | 1.7 | 289 | 2024.07 | No |
> | Diffusion-style | Discrete | VQ-diffusion[7] | 370M | 11.89 | - | 2022.03 | No |
> | Diffusion-style | Discrete | VQ-diffusion-V2[8] | 370M | 7.65 | - | 2023.02 |  |
> | Language-style | Discrete | MaskGIT[9] | 177M | 6.18 | 182.1 | 2022.02 | No |
> | Language-style | Discrete | RCG(cond.)[10] | 502M | 3.49 | 215.5 | 2023.12 | No |
> | Language-style | Discrete | MAGVIT-v2[11] | 307M | 1.78 | 319.4 | 2023.04 | No |
> | Language-style | Discrete | TiTok[12] | 287M | 1.97 | 281.8 | 2024.07 | No |
> | Language-style | Discrete | MaskBit[13] | 305M | 1.52 | 328.6 | 2024.09 | No |
> | Language-style | Discrete | VQVAE[14] | 13.5B | 31.11 | 45 | 2019.06 | No |
> | Language-style | Discrete | VQGAN[15] | 1.4B | 5.2 | 175.1 | 2021.07 | No |
> | Language-style | Discrete | RQTran[16] | 3.8B | 3.8 | 323.7 | 2022.03 | No |
> | Language-style | Discrete | VITVQ[17] | 1.7B | 3.04 | 227.4 | 2022.07 | No |
> | Language-style | Discrete | VAR[18] | 310M | 3.3 | 274.4 | 2024.04 | yes |
> | Language-style |  |  | 600M | 2.57 | 302.6 |  |  |
> | Language-style |  |  | 1B | 2.09 | 312.9 |  |  |
> | Language-style |  |  | 2B | 1.92 | 323.1 |  |  |
> | Language-style | Discrete | LlamaGen[19] | 343M | 3.07 | 256.06 | 2024.07 | yes |
> | Language-style |  |  | 775M | 2.62 | 244.1 |  |  |
> | Language-style |  |  | 1.4B | 2.34 | 253.9 |  |  |
> | Language-style |  |  | 3.1B | 2.18 | 263.3 |  |  |
> | Language-style | continuous | MAR[20] | 208M | 2.31 | 281.7 | 2024.07 | yes |
> | Language-style |  |  | 479M | 1.78 | 296 |  |  |
> | Language-style |  |  | 943M | 1.55 | 303.7 |  |  |

---

> ### Author Response · Authors · 2024-11-21
>
> # **Weakness1**
>
> We greatly appreciate your valuable review comments and hope that our response addresses your concerns regarding the statement: "Bit-level scaling laws and the robustness of discrete auto-regressive models seem to be intuitive and straightforward."
>
> Firstly, as shown in table 1 above, **in the field of vision generative models, there has been ongoing debate regarding the use of discrete versus continuous representation spaces (e.g., [17,18,19,20]).** Both approaches have shown strong performance in terms of scaling laws. **This work, however, takes a different perspective by investigating the impact of these representation spaces on the scaling laws in quantized models.** We find that, despite achieving comparable performance at full precision, discrete autoregressive models consistently outperform continuous models across various quantization settings. **To validate the effectiveness and broad applicability of our conclusions for you, we conducted the same experiments on other models, as detailed in Appendix C. This indicates that our work provides general guidance for subsequent model design and applications in specific deployment scenarios (e.g., mobile devices, edge computing).**
>
> Secondly, while low-bit precision representation often focuses on trading performance for efficiency, **this work demonstrates that by optimizing either the model or quantization algorithm, models can achieve superior bit-level scaling laws. This outstanding characteristic enables the use of lower bit precision to increase model parameters, ultimately enhancing generative capability without sacrificing efficiency.**
>
> **To validate the effectiveness and broad applicability of our conclusions for you, we conducted the same experiments on other models, as detailed in Appendix C. It can be observed that due to the influence of the continuous representation space, MAR, despite exhibiting excellent scaling laws,similar to DiT, do not demonstrate superior bit-level scaling laws. In contrast, LLaMaGen, which shares the discrete representation space with VAR, exhibits outstanding bit-level scaling laws.**
>
> This work provides a deeper, foundational understanding of bit-level scaling laws in visual generative models, from both the model design and quantization algorithm perspectives, supported by rigorous experimental design.
>
> **Our study is an essential step toward understanding how various models and quantization methods influence bit-level scaling behavior, and it also provides the following recommendations for future work:**
>
> We hope the reviewer will take into account the contributions of this work to model design and the application of quantization algorithms. Thank you again!

---

> ### Author Response · Authors · 2024-11-21
>
> # **Question 1**
>
> Thank you very much for your suggestion. In this paper, we have provided additional experiments to demonstrate the effectiveness of TopKLD, as shown in the table below.
>
> |  |  | d16 | d20 | d24 | d30 |
> |---|---|:---:|:---:|:---:|:---:|
> | W16A16 | FP16 | 3.3 | 2.57 | 2.19 | 1.92 |
> | W8A16 | GPTQ | 3.41 | 2.66 | 2.12 | 1.97 |
> | W8A16 | GPTVQ | 3.40 | 2.637 | 2.398 | 2.11 |
> | W8A16 | OmniQ | 3.62 | 2.72 | 2.2098 | 2.0636 |
> | W8A16 | MSE |3.55 |2.71 |2.35 |2.05 |
> | W8A16 | JS Divergence| 3.50| 2.69|2.22 | 2.05|
> | W8A16 | Forward-KLD | 3.41 | 2.636 | 2.40 | 2.05 |
> | W8A16 | Reverse-KLD | 3.41 | 2.636 | 2.41 | 2.04 |
> | W8A16 | TopKLD | 3.40 | 2.634 | 2.394 | 2.01 |
> | W4A16 | GPTQ | 4.64 | 3.247 | 2.572 | 2.277 |
> | W4A16 | GPTVQ | 3.92 | 2.96 | 2.634 | 2.226 |
> | W4A16 | OmniQ | 4.08 | 3.17 | 2.56 | 2.55 |
> | W4A16 | MSE | 3.97| 3.12|2.69 |2.25 |
> | W4A16 | JS Divergence| 3.92 | 3.01 | 2.65 | 2.23 |
> | W4A16 | Forward-KLD | 3.95 | 3.06 | 2.63 | 2.21 |
> | W4A16 | Reverse-KLD | 3.89 | 3.05 | 2.59 | 2.18 |
> | W4A16 | TopKLD | 3.82 | 2.95 | 2.53 | 2.12 |
> | W3A16 | GPTQ | 27.75 | 16.11 | 15.45 | 13.48 |
> | W3A16 | GPTVQ | 12.69 | 9.01 | 6.29 | 5.52 |
> | W3A16 | OmniQ | 18.18 | 10.67 | 6.15 | 3.93 |
> | W3A16 | MSE |4.56 |3.89 |3.54 |3.01 |
> | W3A16 | JS Divergence| 4.45 | 3.72 | 3.25 | 2.51 |
> | W3A16 | Forward-KLD | 4.27 | 3.45 | 2.96 | 2.55 |
> | W3A16 | Reverse-KLD | 4.02 | 3.25 | 2.91 | 2.55|
> | W3A16 | TopKLD | 3.85 | 3.17 | 2.66 | 2.25 |
>
> # **Question 2**
>
> Thank you very much for your suggestion. To investigate the impact of top-k sampling on the bit-level scaling behavior of the model, we performed ablation experiments using different values of K. The results in the table below show that:
>
> 1.While the choice of K does affect the final generation results to some extent, it does not influence the overall trend of the bit-level scaling laws.
>
> 2.The best performance is achieved when the value of K matches the K used in the Top-K sampling during the model's image generation process.
>
> |  | Method | d16 | d20 | d24 | d30 |
> |---|---|---|:---:|---|---|
> | W16A16 | FP | 3.3 | 2.57 | 2.19 | 1.92 |
> | W3A16 | TopKLD(K=400) | 3.95 | 3.21 | 2.77 | 2.29 |
> | W3A16 | TopKLD(K=500) | 3.91 | 3.24 | 2.71 | 2.24 |
> | W3A16 | TopKLD(K=600) | 3.85 | 3.17 | 2.66 | 2.25 |
> | W3A16 | TopKLD(K=700) | 3.92 | 3.19 | 2.72 | 2.25 |
> | W3A16 | TopKLD(K=800) | 3.96 | 3.19 | 2.73 | 2.29 |
>
> # **Question 3**
>
> Thank you very much for your correction. We will revise the error accordingly.

---

> ### Author Response · Authors · 2024-11-21
>
> # **Reference**
>
> [1]Dhariwal P, Nichol A. Diffusion models beat gans on image synthesis[J]. Advances in neural information processing systems, 2021, 34: 8780-8794.
>
> [2] Ho J, Saharia C, Chan W, et al. Cascaded diffusion models for high fidelity image generation[J]. Journal of Machine Learning Research, 2022, 23(47): 1-33.
>
> [3] Rombach R, Blattmann A, Lorenz D, et al. High-resolution image synthesis with latent diffusion models[C]//Proceedings of the IEEE/CVF conference on computer vision and pattern recognition. 2022: 10684-10695.
>
> [4] Peebles W, Xie S. Scalable diffusion models with transformers[C]//Proceedings of the IEEE/CVF International Conference on Computer Vision. 2023: 4195-4205.
>
> [5] Gao S, Zhou P, Cheng M M, et al. Masked diffusion transformer is a strong image synthesizer[C]//Proceedings of the IEEE/CVF International Conference on Computer Vision. 2023: 23164-23173.
>
> [6] Liu Q, Zeng Z, He J, et al. Alleviating Distortion in Image Generation via Multi-Resolution Diffusion Models[J]. arXiv preprint arXiv:2406.09416, 2024.
>
> [7] Gu S, Chen D, Bao J, et al. Vector quantized diffusion model for text-to-image synthesis[C]//Proceedings of the IEEE/CVF conference on computer vision and pattern recognition. 2022: 10696-10706.
>
> [8] Tang Z, Gu S, Bao J, et al. Improved vector quantized diffusion models[J]. arXiv preprint arXiv:2205.16007, 2022.
>
> [9] Chang H, Zhang H, Jiang L, et al. Maskgit: Masked generative image transformer[C]//Proceedings of the IEEE/CVF Conference on Computer Vision and Pattern Recognition. 2022: 11315-11325.
>
> [10] Li T, Katabi D, He K. Self-conditioned image generation via generating representations[J]. arXiv preprint arXiv:2312.03701, 2023.
>
> [11] Yu L, Lezama J, Gundavarapu N B, et al. Language Model Beats Diffusion--Tokenizer is Key to Visual Generation[J]. arXiv preprint arXiv:2310.05737, 2023.
>
> [12] Yu Q, Weber M, Deng X, et al. An Image is Worth 32 Tokens for Reconstruction and Generation[J]. arXiv preprint arXiv:2406.07550, 2024.
>
> [13] Weber M, Yu L, Yu Q, et al. Maskbit: Embedding-free image generation via bit tokens[J]. arXiv preprint arXiv:2409.16211, 2024.
>
> [14] Razavi A, Van den Oord A, Vinyals O. Generating diverse high-fidelity images with vq-vae-2[J]. Advances in neural information processing systems, 2019, 32.
>
> [15] Esser P, Rombach R, Ommer B. Taming transformers for high-resolution image synthesis[C]//Proceedings of the IEEE/CVF conference on computer vision and pattern recognition. 2021: 12873-12883.
>
> [16] Lee D, Kim C, Kim S, et al. Autoregressive image generation using residual quantization[C]//Proceedings of the IEEE/CVF Conference on Computer Vision and Pattern Recognition. 2022: 11523-11532.
>
> [17] Yu J, Li X, Koh J Y, et al. Vector-quantized image modeling with improved vqgan[J]. arXiv preprint arXiv:2110.04627, 2021.
>
> [18] Tian K, Jiang Y, Yuan Z, et al. Visual autoregressive modeling: Scalable image generation via next-scale prediction[J]. arXiv preprint arXiv:2404.02905, 2024.
>
> [19] Sun P, Jiang Y, Chen S, et al. Autoregressive Model Beats Diffusion: Llama for Scalable Image Generation[J]. arXiv preprint arXiv:2406.06525, 2024.
>
> [20] Li, Tianhong, et al. "Autoregressive Image Generation without Vector Quantization." arXiv preprint arXiv:2406.11838 (2024).

---

> ### Author Response · Authors · 2024-11-22
>
> Dear reviewer:
>
> Thanks you for your great efforts in reviewing out paper and providing constructive suggestions/comments. This work focuses on investigating the impact of whether the representation space in vision generation models is continuous or discrete. Furthermore, we propose strategies to optimize bit-level scaling laws under various quantization scenarios. **To address the weaknesses you raised, we have provided extensive examples related to the recent debate on continuous versus discrete representation spaces in vision generation models, as shown in Tab.1. Through numerous experiments presented in Appendix C and Figure 1 in the main text, we demonstrate the significant impact and unique differences of this feature on vision generation models. These findings provide valuable insights into model design and quantization methods, offering guidance for future applications in specific deployment scenarios, such as mobile devices and edge computing.** Additionally, to address your questions 1 and 2 and demonstrate the advantages of TopKLD, we have conducted extensive comparisons with current state-of-the-art methods across various settings, as per your suggestion. If our rebuttal does not address your concerns, you are warmly wecomed to raise questions. If our responses have addressed your concerns, we sincerely request that you consider raising our score.
>
> Best Wishes!
>
> Authors

---

> ### Comment · Reviewer_gPYq · 2024-11-26
>
> Thanks for the author's supplementary experiments. The current results are adequate to illustrate the advantage of TopKLD.
>
> While "Knowledge Distillation in Quantization-Aware Training" (KD-QAT) can enhance VAR's scaling ability at low bits, I am wondering whether KD-QAT also works for DiT.

---

> ### Author Response · Authors · 2024-11-27
>
> Thank you very much for your response!!!
>
> **The following experiments demonstrate that KD-QAT is also effective for DiT.**  Additionally, we have validated its effectiveness for LlamaGen in Appendix C.
>
> However, it is important to note that, compared to discrete representation space models like LlamaGen and VAR, the bit-level scaling laws of DiT are inherently limited by its continuous representation space. If our rebuttal does not address your concerns, you are warmly wecomed to raise questions. If our responses have addressed your concerns, we sincerely request that you consider raising our score.
>
> | Method | #bits | DiT-L/2 | DiT-XL/2 | L-DiT-3 | L-DiT-7 |
> |---|---|---|---|---|---|
> | FP16 | W16A16 | 5.02 | 2.27 | 2.1 | 2.28 |
> | GPTQ | W8A16 | 5.89 | 3.01 | 2.48 | 2.35 |
> | QAT | W8A16 | 5.33 | 2.46 | 2.45 | 2.33 |
> | KD-QAT | W8A16 | 5.15 | 2.32 | 2.26 | 2.27 |
> | GPTQ | W4A16 | 7.8 | 4.52 | 2.56 | 2.31 |
> | QAT | W4A16 | 5.76 | 3.23 | 2.76 | 2.45 |
> | KD-QAT | W4A16 | 5.32 | 3.08 | 2.32 | 2.29 |
> | GPTQ | W3A16 | 32.76 | 25.77 | 12.23 | 14.35 |
> | QAT | W3A16 | 11.23 | 6.34 | 4.76 | 3.78 |
> | KD-QAT | W3A16 | 9.23 | 5.12 | 4.21 | 3.05 |

---

> ### Author Response · Authors · 2024-11-28
>
> Dear reviewer:
>
> Thanks you for providing constructive suggestions. We would like to kindly ask if our responses and additional experiments have addressed all your concerns. If so, we would greatly appreciate it if you could reconsider the score in light of the clarifications and new evidence provided.
>
> Best Wishes!
>
> Authors

---

> ### Author Response · Authors · 2024-11-29
>
> Dear reviewer:
>
> Thanks you for your great efforts in reviewing out paper and providing constructive suggestions/comments. **To address the weaknesses you raised, we have conducted extensive experiments in appendix C and figure 1 of main paper to alleviate concerns.** If our rebuttal does not address your concerns, you are warmly wecomed to raise questions. If our responses have addressed your concerns, we sincerely request that you consider raising our score.
>
> Best Wishes!
>
> Authors

---

### Official Review · Reviewer_gvub · 2024-11-04

**Soundness:** 2
**Presentation:** 3
**Contribution:** 3
**Rating:** 6
**Confidence:** 5

**Summary:**

The paper investigates bit-level scaling laws in quantized vision generative models, specifically comparing diffusion-style and language-style models. The authors find that while both models perform similarly in full precision, language-style models consistently exhibit superior bit-level scaling across various quantization settings. This robustness is attributed to the discrete representation space of language-style models, which enhances resilience to quantization noise. The authors propose TopKLD, a novel knowledge distillation method that balances implicit and explicit knowledge transfer, thereby further optimizing bit-level scaling in quantized models. Their findings provide valuable insights into efficient quantization strategies and underscore the potential of language-style models for low-bit precision applications.

**Strengths:**

1. The paper investigates bit-level scaling laws in quantized vision generative models, specifically comparing diffusion-style and language-style models. The authors find that while both models perform similarly in full precision, language-style models consistently exhibit superior bit-level scaling across various quantization settings. This robustness is attributed to the discrete representation space of language-style models, which enhances resilience to quantization noise.
2. The authors propose TopKLD, a novel knowledge distillation method that balances implicit and explicit knowledge transfer, thereby further optimizing bit-level scaling in quantized models. Their findings provide valuable insights into efficient quantization strategies and underscore the potential of language-style models for low-bit precision applications.

**Weaknesses:**

1. Inconsistent Scaling Comparison in Figure 1: The paper aims to show that language-style models have superior bit-level scaling compared to diffusion-style models. However, the models compared in Figure 1 have different initial total model bits and compute bits, which may itself cause scaling variations. This discrepancy introduces an additional variable that weakens the effectiveness of Figure 1 in supporting the authors’ claim. Aligning initial bit settings could help provide a clearer, more controlled comparison.
2. Limited Advantage of TopKLD in High-Bit Settings: While the authors introduce TopKLD to enhance bit-level scaling, Figure 7(c) and Figure 5(a) suggest that in the W8A8 setting, TopKLD performs similarly to existing methods like SmoothQuant, without a clear improvement. Given that TopKLD introduces extra training overhead, its benefit seems marginal in these high-bit settings. Providing a comparison across a broader range of bit settings could clarify the scenarios where TopKLD is genuinely advantageous.
3. Insufficient Experimental Validation of TopKLD’s Effectiveness: The effectiveness of TopKLD is only partially validated, as shown by its comparison with ForwardKLD and ReverseKLD at 3-bit in Figure 7(b). However, a more comprehensive evaluation against other mainstream quantization methods under varied conditions would provide a stronger basis for its practical effectiveness.
4. Lack of Analysis on the Computational Overhead of TopKLD: TopKLD introduces an additional training overhead, but the paper does not quantify the computational cost compared to existing methods. A detailed analysis of training time, computational resources, and memory requirements would provide a more complete view of its trade-offs, particularly for resource-constrained applications.

**Questions:**

1. Could you provide a more controlled comparison in Figure 1 with equivalent initial model and compute bits for both language-style and diffusion-style models?——The initial bit settings differ between the models, which complicates the interpretation of bit-level scaling behaviors. A more controlled experiment with similar initial bit allocations would strengthen the comparison and isolate the scaling differences more effectively.
2. What specific advantages does TopKLD offer over existing methods in low-bit settings, and could you clarify its computational cost?——While TopKLD is introduced to enhance bit-level scaling, its benefit seems marginal in higher-bit configurations, as shown in Figure 7(c). Could you provide additional data on TopKLD’s performance in low-bit settings and quantify the extra training cost, as well as its memory and computational overhead, compared to other methods like SmoothQuant?
3. Can you expand the experimental validation of TopKLD with comparisons to other mainstream quantization methods across more bit configurations?——The effectiveness of TopKLD is primarily shown in comparison with ForwardKLD and ReverseKLD in the 3-bit setting. Including a broader range of comparisons with other quantization approaches (e.g., OmniQuant, GPTQ) across different bit levels would give a clearer picture of where TopKLD has a distinct advantage.
4. Could you provide additional insights into the potential applications of your findings on bit-level scaling laws?——The study primarily focuses on theoretical scaling improvements, but practical insights or applications for specific deployment scenarios (e.g., mobile devices, edge computing) would make the results more actionable. Could you elaborate on specific scenarios where the bit-level improvements from language-style models might offer a tangible benefit?

---

> ### Author Response · Authors · 2024-11-20
>
> # **Rebuttal Revision Paper Modifications**
>
> We greatly appreciate your valuable review comments. We have revised the paper according to your suggestions and submitted the rebuttal version. For detailed modifications, please refer to the rebuttal version PDF and appendix C: Supplementary materials for rebuttal. Below, we address your identified weaknesses and questions, hoping to resolve your concerns and improve our score.
>
> # **Weakness 1**
>
> Thank you very much for your valuable reminder. We align the initial bit settings to better compare the bit-level scaling laws of language-style models and diffusion-style models. **We have modified Figure 1 in the main paper, as shown in the rebuttal version of the PDF.**
>
> To determine whether a model exhibits superior bit-level scaling laws, we compare the internal trends of the model (e.g., 8-bit VAR vs. 16-bit VAR), rather than making a direct comparison of generative quality between two types of models at the same total bit precision (e.g., 8-bit DiT vs. 8-bit VAR). As shown in Figure 1, **regardless of the quantization method, when full-precision VAR is quantized to lower bit precision, its scaling law shifts towards the lower-left region. In contrast, DiT does not show such a shift.** This is precisely what we mentioned in the caption: "Quantized VAR exhibits better bit-level scaling laws than full-precision VAR, while Quantized DiT shows almost no improvement compared to full precision."
>
> **The characteristics demonstrated by the discrete language-style model allow us to increase the model's parameters through quantization under limited resources, thereby achieving better generative capability.** This is a feature that continuous diffusion-style models lack, and it is precisely what bit-level scaling laws aim to showcase.
>
> # **Weakness 2**
>
> Thank you for your suggestion. We provide a more comprehensive evaluation against other mainstream quantization methods for its practical effectiveness: GPTQ, GPTVQ, SmoothQuant, OmniQuant, and TopKLD. The experiments below demonstrate our superior performance across various bit precisions.
>
> |  |  | d16 | d20 | d24 | d30 |
> |---|---|:---:|:---:|:---:|:---:|
> | W16A16 | FP16 | 3.3 | 2.57 | 2.19 | 1.92 |
> | W8A16 | GPTQ | 3.41 | 2.66 | 2.12 | 1.97 |
> | W8A16 | GPTVQ | 3.40 | 2.637 | 2.398 | 2.11 |
> | W8A16 | OmniQ | 3.62 | 2.72 | 2.2098 | 2.0636 |
> | W8A16 | Forward-KLD | 3.41 | 2.636 | 2.40 | 2.05 |
> | W8A16 | Reverse-KLD | 3.41 | 2.636 | 2.41 | 2.04 |
> | W8A16 | TopKLD | 3.40 | 2.634 | 2.394 | 2.01 |
> | W4A16 | GPTQ | 4.64 | 3.247 | 2.572 | 2.277 |
> | W4A16 | GPTVQ | 3.92 | 2.96 | 2.634 | 2.226 |
> | W4A16 | OmniQ | 4.08 | 3.17 | 2.56 | 2.55 |
> | W4A16 | Forward-KLD | 3.95 | 3.06 | 2.63 | 2.21 |
> | W4A16 | Reverse-KLD | 3.89 | 3.05 | 2.59 | 2.18 |
> | W4A16 | TopKLD | 3.82 | 2.95 | 2.53 | 2.12 |
> | W3A16 | GPTQ | 27.75 | 16.11 | 15.45 | 13.48 |
> | W3A16 | GPTVQ | 12.69 | 9.01 | 6.29 | 5.52 |
> | W3A16 | OmniQ | 18.18 | 10.67 | 6.15 | 3.93 |
> | W3A16 | Forward-KLD | 4.27 | 3.45 | 2.96 | 2.55 |
> | W3A16 | Reverse-KLD | 4.02 | 3.25 | 2.91 | 2.55|
> | W3A16 | TopKLD | 3.85 | 3.17 | 2.66 | 2.25 |
>
> |  | Method | d16 | d20 | d24 | d30 |
> |---|---|---|:---:|---|---|
> | W16A16 | FP | 3.3 | 2.57 | 2.19 | 1.92 |
> | W8A8 | SmoothQ | 3.81 | 2.68 | 2.23 | 2.01 |
> | W8A8 | OmniQ | 3.75 | 2.75 | 2.18 | 2.08 |
> | W8A8 | Forward | 3.8 | 2.72 | 2.16 | 2.10 |
> | W8A8 | TopKLD | 2.75 | 2.7 | 2.18 | 1.98 |
> | W4A8 | SmoothQ | 7.21 | 4.32 | 3.21 | 2.65 |
> | W4A8 | OmniQ | 6.92 | 4.35 | 3.11 | 2.69 |
> | W4A8 | Forward | 6.62 | 3.95 | 3.01 | 2.35 |
> | W4A8 | TopKLD | 5.89 | 3.62 | 2.81 | 2.15 |
>
> As shown, **whether in high-bit or low-bit settings, and whether quantizing only weights or both weights and activations, TopKLD consistently exhibits superior performance.** Even at higher bit precision, TopKLD still leads to noticeable improvements in model accuracy.
>
> Regarding your comment on the “Limited Advantage of TopKLD in High-Bit Settings,” the reason for this is that **our focus is not solely on improving model accuracy but also on scaling laws.** **At high precision levels, models retain sufficient precision, resulting in minimal degradation compared to full-precision models.** Thus, there is no significant enhancement in bit-level scaling in these settings.
>
> **Through the explanation in Weakness1, we believe you can see that models with excellent bit-level scaling laws demonstrate enhanced capabilities at low-bit conditions.** The goal of TopKLD is to improve the scaling ability of models under low-bit conditions. As shown in the results in Section 3.3 of the paper, **TopKLD enhances the bit-level scaling behaviors of language-style models by one level.**

---

> ### Author Response · Authors · 2024-11-20
>
> # **Weakness 3**
>
> Thank you very much for your suggestion. We have provided a comparison of TopKLD with the current mainstream quantization methods. Please refer to the results in Weakness2 for further details.
>
> # **Weakness 4**
>
> Thank you very much for your suggestion. TopKLD is an optimization of current mainstream distillation loss functions. It balances the "implicit knowledge" and "explicit knowledge" derived from full-precision models, thereby enhancing the bit-level scaling behaviors of language-style models by one level. As a result, it does not incur any additional resource overhead compared to methods like ForwardKLD. For your reference, we have provided the training times for TopKLD on  A100 GPU below:
>
> | d16 | d20 | d24 | d30 |
> |---|---|:---:|---|
> | 5.1 | 8.9 | 13.6 | 21.2 |

---

> ### Author Response · Authors · 2024-11-20
>
> # **Questinon 1**
>
> Thank you very much for your reminder. We have used similar initial bit allocations to strengthen the comparison. Please refer to the details in Weakness1.
>
> # **Question 2**
>
> **When a model exhibits excellent bit-level scaling laws, by leveraging this outstanding feature, we can increase the model parameters under limited resources (e.g., in specific deployment scenarios such as mobile devices or edge computing) while maintaining efficiency, ultimately improving generative capabilities.** However, as shown in Figure 5 of the main text, **existing methods fail to achieve better bit-level scaling laws under low-bit settings, which hinders further enhancement of model capabilities in specific deployment scenarios. If you wish to further improve model generation quality under limited resource conditions, TopKLD is an excellent choice.** Although it incurs some additional computational cost, it results in a significant improvement in model performance.
>
> # **Question 3**
>
> Thank you for your valuable suggestion. **We have provided a comparison with existing mainstream quantization methods in Weakness2.** As shown in Figure 5 of the main text, existing methods fail to achieve better bit-level scaling laws under low-bit settings, which hinders further enhancement of model capabilities in specific deployment scenarios (such as mobile devices or edge computing). TopKLD addresses this issue by balancing the "implicit knowledge" and "explicit knowledge" derived from full-precision models, enhancing the bit-level scaling behaviors of language-style models by one level.
>
> # **Question 4**
>
> 1.Potential of Bit Scaling Laws: As shown in Weakness 1, if a model or quantization algorithm is optimized to achieve excellent bit-level scaling laws, it is possible to increase model parameters using lower bit precision while maintaining better generative capability under current resource constraints. **This outstanding feature plays a significant role in specific deployment scenarios, such as mobile devices and edge computing.**
>
> 2.Insights for Model Design: **In the field of vision generative models, there has been ongoing debate regarding the use of discrete versus continuous representation spaces (e.g., [1,2,3,4]).** Both approaches have shown strong performance in terms of scaling laws. This work, however, takes a different perspective by investigating the impact of these representation spaces on the scaling laws in quantized models. **We find that, despite achieving comparable performance at full precision, discrete autoregressive models consistently outperform continuous models across various quantization settings.**
>
> 3.Introduction of a New method: We introduced the TopKLD method, which enhances knowledge transfer from full-precision models by effectively balancing explicit and implicit knowledge, thereby improving the bit-level scaling performance of language-style models.
>
> [1]Li T, Tian Y, Li H, et al. Autoregressive Image Generation without Vector Quantization[J]. arXiv preprint arXiv:2406.11838, 2024.
>
> [2]Tian K, Jiang Y, Yuan Z, et al. Visual autoregressive modeling: Scalable image generation via next-scale prediction[J]. arXiv preprint arXiv:2404.02905, 2024.
>
> [3]Peebles W, Xie S. Scalable diffusion models with transformers[C]//Proceedings of the IEEE/CVF International Conference on Computer Vision. 2023: 4195-4205.
>
> [4]Sun P, Jiang Y, Chen S, et al. Autoregressive Model Beats Diffusion: Llama for Scalable Image Generation[J]. arXiv preprint arXiv:2406.06525, 2024.

---

### Official Review · Reviewer_QXeS · 2024-11-05

**Soundness:** 3
**Presentation:** 2
**Contribution:** 2
**Rating:** 3
**Confidence:** 3

**Summary:**

- This paper presents a systemic analysis of the impact of quantization on vision generative models, particularly comparing diffusion-style and language-style models. Under the bit-level scaling law that has been studied in language modeling, the authors show that the language-style model consistently outperforms the diffusion-style model.

 - The authors also provide explanations and investigations into the reason for their distinctive behaviors in low-bits.

 - To further enhance the bit-level scaling of language-style models, the TopKLD-based distillation method is proposed by balancing implicit knowledge and explicit knowledge.

**Strengths:**

- The paper provides a comprehensive study of how quantization affects two major paradigms of vision generative models, which is crucial for deploying these models efficiently. The finding that language-style models have superior bit-level scaling laws compared to diffusion-style models, might also shed light on further model optimization and deployment.

 -  The proposed TopKLD method for knowledge distillation during the quantization process is innovative and shows experimental promise in improving bit-level scaling laws.

**Weaknesses:**

- The major weakness of this work is the limited scoop. As both VAR and DiT are specific cases in diffusion and language-style vision generative models, their behavior may not apply to other types of vision generative models. Compared to the original paper about k-bit inference scaling laws, the model scope is relatively small, which makes the conclusion unclear to generalize to different model types.

 - The authors provide some analysis about the reason behind models' scaling behaviors and discuss the relevance of the discrete representation. However, vision AR and diffusion models are not distinctive from the representation side. (see question) fds

**Questions:**

- The authors should consider adding different model types into the investigations, that cover more typical language-style and diffusion-style vision generative models.

 - Language-style vision generative models follow the autoregressive modeling in language modeling, while not necessarily being discrete. Similarly, diffusion-style models do not always adopt a continuous representation. How would the analysis apply to discrete diffusion and
continuous AR?

 - Meanwhile, the error analysis from the discrete and continuous domains does not seem to conclude for language-style and diffusion-style models (related to Q2)

---

> ### Author Response · Authors · 2024-11-20
>
> # **Rebuttal Revision Paper Modifications**
>
> We greatly appreciate your valuable review comments. We have revised the paper according to your suggestions and submitted the rebuttal version. **For detailed modifications, please refer to the rebuttal version PDF and appendix C: Supplementary materials for rebuttal.** Below, we address your identified weaknesses and questions, hoping to resolve your concerns and improve our score.
>
> # **Table 1**
>
> | Model   Type | Discrete/Continuous | model | #para | FID | IS | dates | Scaling ability |
> |:---:|:---:|:---:|:---:|:---:|:---:|:---:|:---:|
> | Diffusion-style | continuous | ADM[1] | 554M | 10.94 | 101 | 2021.07 | No |
> | Diffusion-style | continuous | CDM[2] | - | 4.88 | 158.7 | 2021.12 | No |
> | Diffusion-style | continuous | LDM-8[3] | 258M | 7.76 | 209.5 | 2022.04 | No |
> | Diffusion-style | continuous | LDM-4 | 400M | 3.6 | 247.7 |  | No |
> | Diffusion-style | continuous | DiT[4] | 458M | 5.02 | 167.2 | 2023.03 | Yes |
> | Diffusion-style |  |  | 675M | 2.27 | 278.2 |  |  |
> | Diffusion-style |  |  | 3B | 2.1 | 304.4 |  |  |
> | Diffusion-style |  |  | 7B | 2.28 | 316.2 |  |  |
> | Diffusion-style | continuous | MDTv[5] | 676M | 1.58 | 314.7 | 2024.02 | No |
> | Diffusion-style | continuous | DiMR[6] | 505M | 1.7 | 289 | 2024.07 | No |
> | Diffusion-style | Discrete | VQ-diffusion[7] | 370M | 11.89 | - | 2022.03 | No |
> | Diffusion-style | Discrete | VQ-diffusion-V2[8] | 370M | 7.65 | - | 2023.02 |  |
> | Language-style | Discrete | MaskGIT[9] | 177M | 6.18 | 182.1 | 2022.02 | No |
> | Language-style | Discrete | RCG(cond.)[10] | 502M | 3.49 | 215.5 | 2023.12 | No |
> | Language-style | Discrete | MAGVIT-v2[11] | 307M | 1.78 | 319.4 | 2023.04 | No |
> | Language-style | Discrete | TiTok[12] | 287M | 1.97 | 281.8 | 2024.07 | No |
> | Language-style | Discrete | MaskBit[13] | 305M | 1.52 | 328.6 | 2024.09 | No |
> | Language-style | Discrete | VQVAE[14] | 13.5B | 31.11 | 45 | 2019.06 | No |
> | Language-style | Discrete | VQGAN[15] | 1.4B | 5.2 | 175.1 | 2021.07 | No |
> | Language-style | Discrete | RQTran[16] | 3.8B | 3.8 | 323.7 | 2022.03 | No |
> | Language-style | Discrete | VITVQ[17] | 1.7B | 3.04 | 227.4 | 2022.07 | No |
> | Language-style | Discrete | VAR[18] | 310M | 3.3 | 274.4 | 2024.04 | yes |
> | Language-style |  |  | 600M | 2.57 | 302.6 |  |  |
> | Language-style |  |  | 1B | 2.09 | 312.9 |  |  |
> | Language-style |  |  | 2B | 1.92 | 323.1 |  |  |
> | Language-style | Discrete | LlamaGen[19] | 343M | 3.07 | 256.06 | 2024.07 | yes |
> | Language-style |  |  | 775M | 2.62 | 244.1 |  |  |
> | Language-style |  |  | 1.4B | 2.34 | 253.9 |  |  |
> | Language-style |  |  | 3.1B | 2.18 | 263.3 |  |  |
> | Language-style | continuous | MAR[20] | 208M | 2.31 | 281.7 | 2024.07 | yes |
> | Language-style |  |  | 479M | 1.78 | 296 |  |  |
> | Language-style |  |  | 943M | 1.55 | 303.7 |  |  |

---

> ### Author Response · Authors · 2024-11-20
>
> # **Weakness 1**
>
> We greatly appreciate your valuable review comments. As shown in Table 1, the field of vision generation models currently has two main development paths, and some of these paths exhibit excellent scaling laws. This paper is based on models that, at the time of our research, had already demonstrated scaling laws, which are DiT and VAR, for further exploration.
>
> Due to the chronological order of submissions, scaling laws have also recently been observed in the continuous autoregressive model domain, specifically in MAR. Therefore, following your suggestion, we conducted the same experiments to verify the correctness of our conclusions. Additionally, we validated our findings on LlamaGen, a model similar to VAR, to further enhance the generalizability of our conclusions.
>
> **The results of these experiments can be found in Appendix C2 of the rebuttal revision, titled "Empirical Validation Through Additional Models."**
>
> # **Weakness 2**
>
> From Table 1, **we can observe that in the field of vision generative models, there has been ongoing debate regarding the use of discrete versus continuous representation spaces.[20]** Both approaches have shown strong performance in terms of scaling laws. **This work, however, takes a different perspective by investigating the impact of these representation spaces on the scaling laws in quantized models.** We find that, despite achieving comparable performance at full precision, discrete  models consistently outperform continuous models across various quantization settings.  **Additionally, through our additional experiments on continuous autoregressive models and discrete AR models in Appendix C, as well as the analysis in Section 3.2, it becomes evident that the nature of the representation space—discrete or continuous—has a significant impact on determining whether AR and diffusion models exhibit superior bit-level scaling laws.**
>
> # **Question1**
>
> We conducted a statistical summary of vision generation models and, based on this analysis, selected models that have already reported scaling laws for further exploration: MAR[20], VAR[18], DiT[4], and LlamaGen[19]. The details of this overview can be found in Appendix C1 of the rebuttal revision, titled "Overview."
>
> # **Question 2**
>
> As shown in Table 1, current discrete diffusion models do not exhibit scaling laws, making it impossible to explore their bit-level scaling laws. Therefore, **we focused on supplementary research into continuous autoregressive models. The results show that, due to their continuous representation space, the bit-level scaling laws of continuous autoregressive models are not as strong as those observed in discrete models, which aligns with our conclusions.**
>
> At the same time, **we observed that LlamaGen, a discrete autoregressive model, demonstrates the same excellent bit-level scaling laws. This suggests that the observed scaling behavior is not specific to VAR but is instead a result of the discrete representation space, as discussed in Section 3.2.** Since the representation space has been abstracted, this characteristic holds universally across various discrete models, as detailed in Section 3.2.
>
> # **Question 3**
>
> Thank you very much for your valuable suggestions. In the field of vision generative models, **there has been ongoing debate regarding the use of discrete versus continuous representation spaces (e.g., [17,18,19,20]).** Both approaches have shown strong performance in terms of scaling laws. **This work, however, takes a different perspective by investigating the impact of these representation spaces on the scaling laws in quantized models.** We find that, despite achieving comparable performance at full precision, discrete autoregressive models consistently outperform continuous models across various quantization settings.
>
> **Our study is an essential step toward understanding how various models and quantization methods influence bit-level scaling behavior, and it also provides the following recommendations for future work:**
>
> From our exploration, we can conclude that **discrete representation space reconstruction offers a more stable foundation for scaling at low bit precision.** Moreover, we introduced the TopKLD method, which enhances knowledge transfer from full-precision models by effectively balancing explicit and implicit knowledge, thereby improving bit-level scaling performance. This study indicates that achieving optimal bit-level scaling behavior requires a synergistic interaction between model design and quantization algorithms.

---

> ### Author Response · Authors · 2024-11-20
>
> # **Reference**
>
> [1]Dhariwal P, Nichol A. Diffusion models beat gans on image synthesis[J]. Advances in neural information processing systems, 2021, 34: 8780-8794.
>
> [2] Ho J, Saharia C, Chan W, et al. Cascaded diffusion models for high fidelity image generation[J]. Journal of Machine Learning Research, 2022, 23(47): 1-33.
>
> [3] Rombach R, Blattmann A, Lorenz D, et al. High-resolution image synthesis with latent diffusion models[C]//Proceedings of the IEEE/CVF conference on computer vision and pattern recognition. 2022: 10684-10695.
>
> [4] Peebles W, Xie S. Scalable diffusion models with transformers[C]//Proceedings of the IEEE/CVF International Conference on Computer Vision. 2023: 4195-4205.
>
> [5] Gao S, Zhou P, Cheng M M, et al. Masked diffusion transformer is a strong image synthesizer[C]//Proceedings of the IEEE/CVF International Conference on Computer Vision. 2023: 23164-23173.
>
> [6] Liu Q, Zeng Z, He J, et al. Alleviating Distortion in Image Generation via Multi-Resolution Diffusion Models[J]. arXiv preprint arXiv:2406.09416, 2024.
>
> [7] Gu S, Chen D, Bao J, et al. Vector quantized diffusion model for text-to-image synthesis[C]//Proceedings of the IEEE/CVF conference on computer vision and pattern recognition. 2022: 10696-10706.
>
> [8] Tang Z, Gu S, Bao J, et al. Improved vector quantized diffusion models[J]. arXiv preprint arXiv:2205.16007, 2022.
>
> [9] Chang H, Zhang H, Jiang L, et al. Maskgit: Masked generative image transformer[C]//Proceedings of the IEEE/CVF Conference on Computer Vision and Pattern Recognition. 2022: 11315-11325.
>
> [10] Li T, Katabi D, He K. Self-conditioned image generation via generating representations[J]. arXiv preprint arXiv:2312.03701, 2023.
>
> [11] Yu L, Lezama J, Gundavarapu N B, et al. Language Model Beats Diffusion--Tokenizer is Key to Visual Generation[J]. arXiv preprint arXiv:2310.05737, 2023.
>
> [12] Yu Q, Weber M, Deng X, et al. An Image is Worth 32 Tokens for Reconstruction and Generation[J]. arXiv preprint arXiv:2406.07550, 2024.
>
> [13] Weber M, Yu L, Yu Q, et al. Maskbit: Embedding-free image generation via bit tokens[J]. arXiv preprint arXiv:2409.16211, 2024.
>
> [14] Razavi A, Van den Oord A, Vinyals O. Generating diverse high-fidelity images with vq-vae-2[J]. Advances in neural information processing systems, 2019, 32.
>
> [15] Esser P, Rombach R, Ommer B. Taming transformers for high-resolution image synthesis[C]//Proceedings of the IEEE/CVF conference on computer vision and pattern recognition. 2021: 12873-12883.
>
> [16] Lee D, Kim C, Kim S, et al. Autoregressive image generation using residual quantization[C]//Proceedings of the IEEE/CVF Conference on Computer Vision and Pattern Recognition. 2022: 11523-11532.
>
> [17] Yu J, Li X, Koh J Y, et al. Vector-quantized image modeling with improved vqgan[J]. arXiv preprint arXiv:2110.04627, 2021.
>
> [18] Tian K, Jiang Y, Yuan Z, et al. Visual autoregressive modeling: Scalable image generation via next-scale prediction[J]. arXiv preprint arXiv:2404.02905, 2024.
>
> [19] Sun P, Jiang Y, Chen S, et al. Autoregressive Model Beats Diffusion: Llama for Scalable Image Generation[J]. arXiv preprint arXiv:2406.06525, 2024.
>
> [20] Li, Tianhong, et al. "Autoregressive Image Generation without Vector Quantization." arXiv preprint arXiv:2406.11838 (2024).

---

> ### Author Response · Authors · 2024-11-22
>
> Dear reviewer:
>
> Thanks you for your great efforts in reviewing out paper and providing constructive suggestions/comments. **To address the weakness you raised regarding the limited scope, we have provided extensive examples of the recent debate on continuous versus discrete representation spaces in vision generation models, as presented in Table 1. Additionally, following your suggestion, we conducted numerous experiments detailed in Appendix C and Figure 1 of the main text to demonstrate the validity and generalizability of our conclusions on bit-level scaling laws for vision generation models. These experiments cover various mainstream research directions in vision generation models.** If our rebuttal does not address your concerns, you are warmly wecomed to raise questions. If our responses have addressed your concerns, we sincerely request that you consider raising our score.
>
> Best Wishes!
>
> Authors

---

> ### Author Response · Authors · 2024-11-27
>
> Dear reviewer:
>
> Thanks you for your great efforts in reviewing out paper and providing constructive suggestions/comments. **To address the weaknesses you raised, we have conducted extensive experiments in appendix C and figure 1 of main paper to alleviate concerns.** If our rebuttal does not address your concerns, you are warmly wecomed to raise questions. If our responses have addressed your concerns, we sincerely request that you consider raising our score.
>
> Best Wishes!
>
> Authors

---

> ### Author Response · Authors · 2024-11-29
>
> Dear reviewer:
>
> Thanks you for your great efforts in reviewing out paper and providing constructive suggestions/comments. **To address the weaknesses you raised, we have conducted extensive experiments in appendix C and figure 1 of main paper to alleviate concerns.** If our rebuttal does not address your concerns, you are warmly wecomed to raise questions. If our responses have addressed your concerns, we sincerely request that you consider raising our score.
>
> Best Wishes!
>
> Authors

---

### Meta-Review · Area_Chair_gma9 · 2024-12-19

**Metareview:**

The paper got mostly negative ratings. The reviewers cited limited scope, insufficient experimental evaluation, lack of computational overhead analysis. They also raised a number of questions. The authors tried to address the concerns during the discussion period and provided a lot of additional evaluations and details. Reviewers unfortunately were not engaged during this period, with an exception, and the scores didn't improve. The AC believes the paper didn't find enough support from the community. The authors went further and wrote a message to ACs and PCs, in which they explained their concerns about proper evaluation of their manuscript. The AC went through the reviews, responses, message to AC, looked through the paper. AC believes that while the reviewers could have been more responsive indeed, the number of issues they raised clearly shows that the paper didn't get enough traction with the community. And hence the decision.

**Additional Comments On Reviewer Discussion:**

There was no extensive discussion between reviewers and authors, which is uncommon for ICLR.

---

### Decision · Program_Chairs · 2025-01-22

Reject